The EMBO Journal (2013) **32**, 2561–2574
www.embojournal.org

# Identification of the missing pluripotency mediator downstream of leukaemia inhibitory factor

## Graziano Martello[1,*], Paul Bertone[1,2,3] and Austin Smith[1,4,*]

[1]Wellcome Trust—Medical Research Council Cambridge Stem Cell Institute, University of Cambridge, Cambridge, UK, [2]European Molecular Biology Laboratory, European Bioinformatics Institute, Cambridge, UK, [3]Genome Biology and Developmental Biology Units, European Molecular Biology Laboratory, Heidelberg, Germany and [4]Department of Biochemistry, University of Cambridge, Cambridge, UK

**Self-renewal of pluripotent mouse embryonic stem (ES) cells is sustained by the cytokine leukaemia inhibitory factor (LIF) acting through the transcription factor Stat3. Several targets of Stat3 have previously been identified, most notably the reprogramming factor Klf4. However, such factors are neither required nor sufficient for the potent effect of LIF. We took advantage of *Stat3* null ES cells to confirm that Stat3 mediates the self-renewal response to LIF. Through comparative transcriptome analysis intersected with genome location data, we arrived at a set of candidate transcription factor effectors. Among these, Tfcp2l1 (also known as Crtr-1) was most abundant. Constitutive expression of Tfcp2l1 at levels similar to those induced by LIF effectively substituted for LIF or Stat3 in sustaining clonal self-renewal and pluripotency. Conversely, knockdown of Tfcp2l1 profoundly compromised responsiveness to LIF. We further found that Tfcp2l1 is both necessary and sufficient to direct molecular reprogramming of post-implantation epiblast stem cells to naïve pluripotency. These results establish Tfcp2l1 as the principal bridge between LIF/Stat3 input and the transcription factor core of naïve pluripotency.**

*The EMBO Journal* (2013) **32**, 2561–2574. doi:10.1038/emboj.2013.177; Published online 13 August 2013
*Subject Categories:* signal transduction; development
*Keywords:* ES cell self-renewal; LIF; pluripotency; reprogramming

## Introduction

Early mammalian embryos are characterized by the presence of a regulative population of cells each with the capability of giving rise to all somatic lineages and to germ cells. This property, pluripotency, first emerges in a naïve form in the epiblast in the pre-implantation blastocyst (Nichols and Smith, 2012). When exposed to an appropriate environment

*ex vivo*, mouse naïve epiblast cells can be expanded indefinitely as embryonic stem (ES) cells (Evans and Kaufman, 1981; Martin, 1981; Nichols *et al*, 1990; Brook and Gardner, 1997). The cytokine leukaemia inhibitory factor (LIF) potently promotes ES cell self-renewal (Smith *et al*, 1988; Williams *et al*, 1988), and is routinely used in the derivation and culture of mouse ES cells (Smith, 2001). Binding of LIF to the gp130/LIF-R complex leads to activation of JAK kinases (Yoshida *et al*, 1994) that, in turn, phosphorylate the transcription factor Stat3 (Akira *et al*, 1994; Boeuf *et al*, 1997). Phosphorylated Stat3 dimerizes, enters the nucleus, and activates the expression of target genes (Zhong *et al*, 1994; Boeuf *et al*, 1997; Burdon *et al*, 2002; Bourillot *et al*, 2009). Artificial activation of Stat3 is sufficient to sustain ES cell self-renewal in the absence of LIF (Burdon *et al*, 1999; Matsuda *et al*, 1999), whereas antagonism of Stat3 leads to differentiation (Niwa *et al*, 1998; Bourillot *et al*, 2009). These findings indicate that Stat3 is the key mediator of LIF action in ES cells. However, LIF activates via JAK other signalling pathways that have also been proposed to play a role in ES cell maintenance (Welham *et al*, 2007; Niwa *et al*, 2009; Griffiths *et al*, 2011).

LIF/Stat3 signalling also plays a key role in the conversion of primordial germ cells into pluripotent EG cells (Matsui *et al*, 1992; Resnick *et al*, 1992; Leitch *et al*, 2013) and facilitates transcription factor directed reprogramming. Hyperactivation of Stat3 potently enhances reprogramming of somatic cells into induced pluripotent stem (iPS) cells (van Oosten *et al*, 2012), while blockade of the LIF/Stat3 pathway abolishes iPS cell generation (Tang *et al*, 2012). Furthermore, post-implantation epiblast stem cells, EpiSCs (Brons *et al*, 2007; Tesar *et al*, 2007), can be reprogrammed to a naïve pluripotent state simply by transient activation of Stat3 (Yang *et al*, 2010). These actions are separable from the self-renewal effect of LIF/Stat3 on established pluripotent stem cells (Yang *et al*, 2010).

Despite the central role ascribed to Stat3 in mouse pluripotency, its downstream effectors are incompletely described. LIF does not directly regulate core pluripotency factors Oct4, Sox2, Nanog, or Esrrb. Efforts to delineate the transcriptional programme stimulated by LIF have identified transcription factor targets of Stat3, such as *Klf4*, *Pim1*, and *Gbx2* (Li *et al*, 2005; Hall *et al*, 2009; Niwa *et al*, 2009; Tai and Ying, 2013). However, none of these factors are indispensable for LIF responsiveness, nor can their forced expression fully recapitulate LIF activity. Notably, Klf4 is one of the four canonical Yamanaka factors that direct somatic cell reprogramming (Takahashi and Yamanaka, 2006), but LIF is required in addition (Tang *et al*, 2012). These observations suggest either, functional redundancy and additive effects between multiple Stat3 targets, or alternatively the existence of a pivotal unidentified target.

We previously showed that two selective small molecule inhibitors (2i) of Gsk3 and Mek kinases eliminate ES cell differentiation and can sustain self-renewal in the absence of LIF (Ying *et al*, 2008; Wray *et al*, 2010). Furthermore,

*Corresponding authors. G Martello, Wellcome Trust—Medical Research Council Stem Cell Institute, Tennis Court Road, Cambridge CB2 1QR, UK. Tel.: +44 (0) 1223 760 281; Fax: +44 (0) 1223 760 241; E-mail: gm419@cam.ac.uk or A Smith, Wellcome Trust—Medical Research Council Stem Cell Institute, University of Cambridge, Tennis Court Road, Cambridge CB2 1QR, UK. Tel.: +44 (0) 1223 760 233; Fax: +44 (0) 1223 760 241; E-mail: austin.smith@cscr.cam.ac.uk

2i allows derivation and expansion of $Stat3^{-/-}$ ES cells. Here, we exploited these null cells in a refined search for critical effectors of the LIF/Stat3 pathway.

## Results

### Loss of Stat3 has no effect on potency of ES cells

*Stat3* null ES cells can be derived and expanded when differentiation stimuli are blocked using 2i (Ying *et al*, 2008). We confirmed the identity and pluripotency of these cells by chimaera formation after blastocyst injection (Figure 1A). Consistent with this, when we examined the expression of genes associated with either pluripotency or germ layer specification we found no major differences between *Stat3* null and wild-type cells maintained in 2i (Figure 1B). Furthermore, null cells did not exhibit any overt sign of spontaneous differentiation or appreciable cell death (Figure 1C) and were able to generate undifferentiated colonies at clonal density with efficiency equal to wild-type cells (Figure 1D). We therefore conclude that deletion of *Stat3* does not impair ES cell self-renewal efficiency in 2i. In other culture conditions, however, the mutant cells cannot

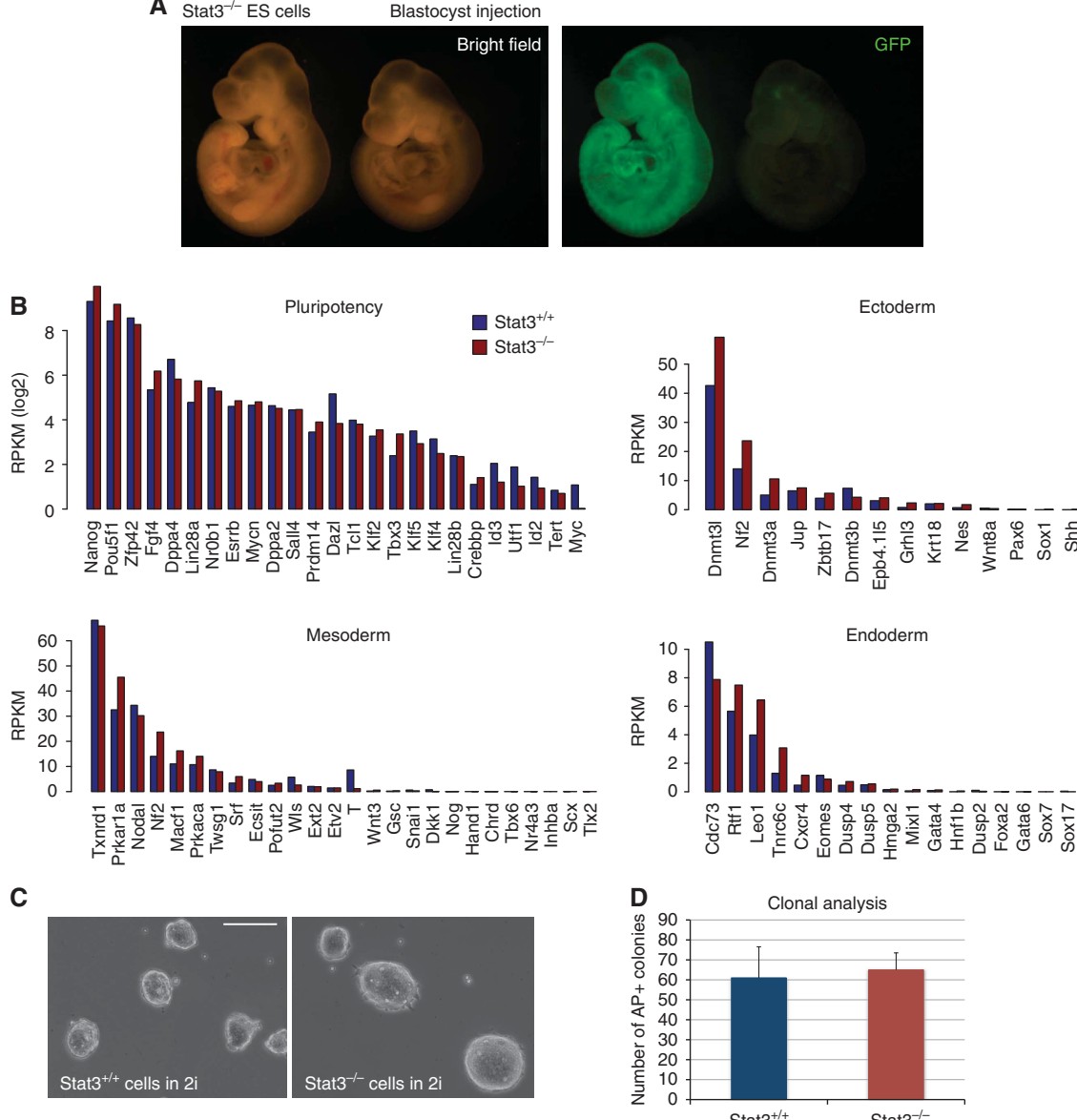

**Figure 1** Absence of Stat3 does not alter ES cell identity, pluripotency, or self-renewal in 2i. (**A**) GFP-labelled $Stat3^{-/-}$ ES cells were injected into blastocyst stage embryos; embryos were scored at E9.5 for the presence of GFP-positive cells and 15 embryos out of 19 showed widespread chimaerism. At mid-gestation (E12.5), 3 out of 3 embryos showed widespread chimaerism (not shown). A representative chimaeric embryo at E9.5 is shown. The embryo on the right showed no chimaerism and serves as a control for autofluorescence. See also Supplementary Figure S1A. (**B**) Absolute expression levels measured by RNA sequencing of genes associated with pluripotency or lineage priming, in $Stat3^{+/+}$ and $Stat3^{-/-}$ cells. RPKM (Reads Per Kilobase per Million mapped reads) is a normalized unit of mRNA expression. (**C**) Morphology of $Stat3^{+/+}$ and $Stat3^{-/-}$ cells in 2i culture. Both cell lines show homogeneous morphology with no sign of spontaneous differentiation. Scale bar, 100 μm. (**D**) Clonogenicity assay on Stat3$^{+/+}$ and Stat3$^{-/-}$ cells. Three hundred cells per well were plated in 2i on laminin-coated plates and stained for alkaline phosphatase (AP) after 5 days. Bars show the number of AP-positive colonies obtained. Mean and s.d. of two independent experiments is shown.

self-renew (Ying *et al*, 2008) because they are non-responsive to LIF, indicating that activation of Stat3 cannot be substituted by alternative mediators.

### Identification of Stat3 direct targets in mouse ES cells

Nonetheless, LIF is known to activate PI3 kinase and Erk signalling in addition to Stat3 (Burdon *et al*, 1999). It is also known that LIF enhances self-renewal efficiency in all ES cell culture conditions, including 2i (Wray *et al*, 2010). We therefore tested whether LIF has any additive effect on self-renewal of *Stat3* null cells. As previously observed, LIF increased the colony-forming efficiency of wild-type cells (Figure 2A). However, *Stat3* null cells showed no response, further verifying the primary role of Stat3 in mediating the contribution of LIF to ES cell self-renewal.

The existence and developmental potency of *Stat3* null ES cells highlights the regulative nature of the naïve pluripotency network (Nichols and Smith, 2012). This flexibility creates the opportunity for manipulating the extrinsic environment to delineate the functional contributions of individual components (Martello *et al*, 2012). Accordingly, we exploited these mutant ES cells to define genes that are directly induced by activation of Stat3 rather than other signals downstream of LIF receptor. We exposed wild-type and *Stat3* null cells to LIF for 1 h and prepared RNA for transcriptome analysis by deep sequencing. The short period of LIF stimulation is expected to enrich for primary transcriptional targets. We found that 188 genes were induced in *Stat3* wild-type cells (Figure 2B, orange), and among these only 5 were induced in *Stat3* null cells (Figure 2B, green). This indicates that the majority of genes acutely responsive to LIF require Stat3 for induction. We then used published Stat3 ChIP-seq (chromatin immunoprecipitation followed by massively parallel sequencing) data (Chen *et al*, 2008) to generate a list of genes (see Materials and methods) bound, and thus potentially directly regulated, by Stat3 (top panel of Figure 2B, purple). This yielded 3935 unique genes, representing ~17% of all annotated genes. Significantly, a high proportion of genes induced by LIF in *Stat3* wild-type cells were also bound by Stat3 (38.8%, $P$-value $= 3.4 \times 10^{-14}$). In contrast, Stat3-bound genes were significantly underrepresented among those induced by LIF in *Stat3* null cells (1.9%, $P$-value $= 0.0094$). We performed a similar analysis on genes downregulated after LIF stimulation (bottom panel of Figure 2B), and found that Stat3-bound genes were underrepresented (11.2%, $P$-value $= 0.003$). These results are consistent with the characterized role of Stat3 as a transcriptional activator (Boeuf *et al*, 1997; Darnell, 1997).

Accordingly, we focussed on genes induced by LIF in wild-type cells (Figure 2C and D), and selected those involved in the regulation of transcription, such as transcription factors or chromatin modifiers. We performed gene expression analysis by qRT-PCR on a set of 13 genes and validated 6 to be induced after LIF stimulation (Figure 2D). Induction of the remaining seven genes could not be confirmed because they are expressed at low levels and thus were difficult to quantify reliably by qRT-PCR. Among these six genes are *Stat3* itself and other genes previously identified as Stat3 direct targets (Bourillot *et al*, 2009). Interestingly, however, the most highly expressed transcription factor in this group, *Tfcp2l1* (also known as *Crtr-1*), has not previously been linked to LIF signalling. We therefore tested whether *Tcfp2l1* is indeed

directly responsive to LIF by stimulating cells in the presence of the protein synthesis inhibitor cycloheximide (Figure 2F). Induction was unaffected, confirming that *Tfcp2l1* is a primary target of LIF signal transduction.

We then tested dependence on LIF for sustained expression. For this, we cultured ES cells under serum-free conditions in the presence of LIF and the Mek inhibitor PD0325901 (PD), then withdrew LIF and added a JAK kinase inhibitor (JAKi) to block LIF signalling completely. We analysed the expression of the six Stat3 targets over time and found that, with the exception of *Gbx2*, all genes were downregulated upon removal of the LIF signal (Figure 2G). Expression of *Klf4*, *Pim1*, *Prr13*, *Stat3*, and *Tfcp2l1* therefore directly correlates with LIF stimulation, whereas *Gbx2* expression is responsive to LIF (Figure 2D and E) but can be maintained by additional mechanisms.

### Tfcp2l1 is the major effector of self-renewal downstream of Stat3

LIF, in combination with serum, maintains ES cell self-renewal. Therefore, omitting Stat3 itself, we tested whether the identified targets could confer self-renewal in serum without LIF. We used Rex1GFPd2 reporter cells to monitor uncommitted status (Marks *et al*, 2012; Martello *et al*, 2012) and transfected these with piggyBac expression vectors (Guo *et al*, 2009). Following hygromycin selection for two passages, we tested the response to LIF withdrawal in the presence of serum. Empty vector (PB-Vector)-transfected cells rapidly downregulated the GFP reporter and differentiated (Figure 3A; Supplementary Figure S1B). This was unchanged in *Pim1* and *Prr13* transfectants. *Klf4* and *Gbx2* expression maintained a proportion of ES cells in a morphologically undifferentiated state for several passages, and this was reflected in persistent expression of the Rex1-GFP reporter. However, *Tfcp2l1* transfectants showed a stronger suppression of differentiation and maintenance of Rex1GFP, comparable to continuous culture in LIF (Figure 3A, compare top-left and bottom-right flow profiles). We also examined self-renewal capacity at the single-cell level by colony forming assay (Figure 3B). *Gbx2* and *Klf4* transfectants produced some undifferentiated colonies in the absence of LIF, consistent with previous reports (Hall *et al*, 2009; Niwa *et al*, 2009; Tai and Ying, 2013), but many more colonies were obtained from *Tfcp2l1* transfectants.

*Klf4* and *Gbx2* have previously been identified as factors promoting self-renewal downstream of LIF. However, the preceding data suggest that Tfcp2l1 has more potent self-renewal activity. LIF also supports self-renewal in serum-free conditions when combined with Mek inhibition (PD) (Wray *et al*, 2010). We found that *Tfcp2l1* expression conferred long-term self-renewal capacity under serum-free conditions in the presence of PD without LIF (Figure 3C, see also Figure 4). Thus, overexpression of *Tfcp2l1* largely recapitulates LIF stimulation in different culture conditions.

We then tested which target genes are required for LIF-responsive ES cell self-renewal by knockdown using a doxycycline (DOX)-inducible shRNA system (see Materials and methods). For each gene, we used two independent shRNA constructs with knockdown efficiencies > 60% (Supplementary Figure S1C). When cultured in serum-containing media in the presence of LIF, ES cells transfected with a control shRNA (shLacZ) gave rise to the same number of colonies in the

presence or absence of DOX (Figure 3D, first bar). Knockdown of *Klf4* or *Gbx2* had no effect in agreement with previous reports showing that suppression of either of these factors does not markedly impair ES cell self-renewal

(Jiang *et al*, 2008; Tai and Ying, 2013). In contrast, *Tfcp2l1* knockdown caused a marked decrease in the number of ES cell colonies (Figure 3D). Similar results were obtained in LIF/PD (Figure 3E). Furthermore, *Tfcp2l1* depletion in bulk

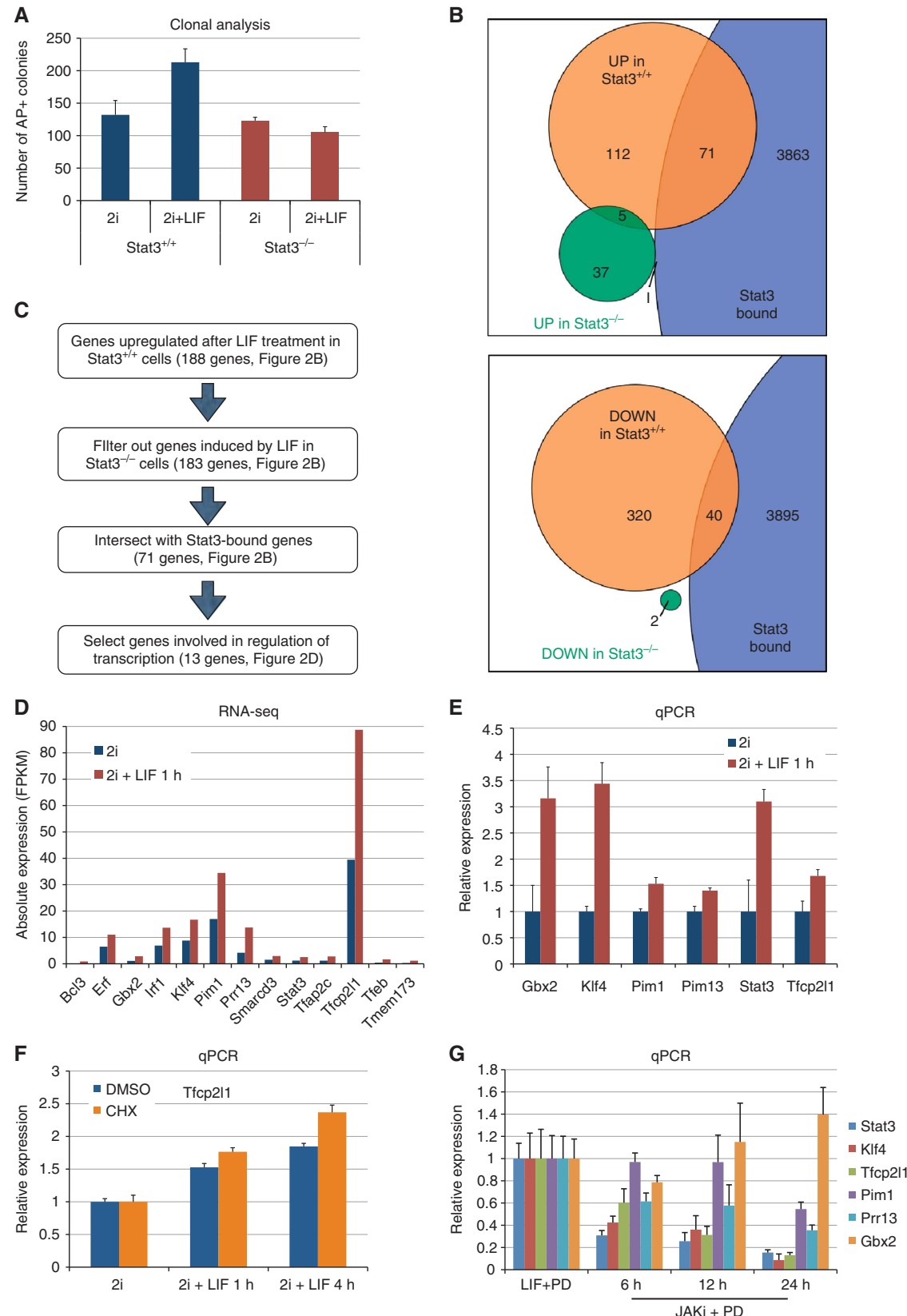

culture in presence of LIF precipitated differentiation (Figure 3F, middle and bottom panels, see also Supplementary Figure S1F). Consistent with this, gene expression analysis after 4 days of shTcfcp2l1 induction showed a reduction in mRNA levels of key pluripotency factors, including *Nanog*, *Oct3/4*, *Sox2*, and *Esrrb* (Figure 3G). Significantly, however, *Tfcp2l1* knockdown was well tolerated by ES cells cultured in 2i + LIF (Supplementary Figure S1D and E), confirming that the phenotype is specific to LIF-stimulated self-renewal. These results suggest that Tfcp2l1 exerts a unique function downstream of Stat3 that is necessary for a full self-renewal response to LIF (Figure 3I).

Many naïve pluripotency factors such as Klf4 exhibit mosaic expression in ES cells cultured in LIF and serum (Wray *et al*, 2010). In contrast, we found that Tfcp2l1 protein is widely expressed, with >90% of Oct3/4 expressing cells being Tfcp2l1 positive (Figure 3H and data not shown). This broad expression may underlie the capacity of LIF to sustain self-renewal efficiently in serum.

As shown above, *Tfcp2l1* overexpression can maintain self-renewal of ES cells in the absence of LIF (Figure 3A–C). In those experiments, however, *Tfcp2l1* transgene mRNA may be up to three-fold higher than endogenous levels (Supplementary Figure S1B) which, given that endogenous *Tfcp2l1* transcript is very abundant in ES cells (Figure 2D), raises the possibility of neomorphic effects. We therefore used low-dose Hygromycin selection to obtain ES cell transfectants in which *Tfcp2l1* transgene expression was restricted at close to LIF-stimulated endogenous levels (Figure 4A), as measured by qRT–PCR and immunoblotting (Figure 4B). This constrained expression of *Tfcp2l1* was fully sufficient to sustain self-renewal after LIF withdrawal, as shown by undifferentiated morphology (Figure 4C), expression of pluripotency markers (Figure 4D and F), and capacity to form AP-positive colonies (Figure 4E).

We then tested whether the effect of Tfcp2l1 on self-renewal in these cells was reversible by excising the floxed *Tfcp2l1* transgene after expansion in the absence of LIF for seven passages (Figure 4A; Supplementary Figure S1G). After excision, cells differentiated unless maintained in the presence of LIF (Figure 4C–E). We carried out blastocyst injection with the excised cells to assay their developmental identify and potency. As a rigorous test, we injected single cells. We obtained chimaeras with an efficiency of ~30%, similar to the parental ES cell line (Figure 4G, table). Mid-gestation chimaeras exhibited widespread contribution to all germ layers (Figure 4G, top panels). Single *Tfcp2l1* excised cells could also give rise to healthy term chimaeras (Figure 4H). We conclude that constitutive expression of *Tfcp2l1* is sufficient to replace LIF and sustains ES cell self-renewal without transformation or impairment of somatic differentiation potential. Collectively, these results indicate that Tfcp2l1 is the major effector of self-renewal downstream of Stat3.

### Tfcp2l1 sustains self-renewal independently of Stat3

Next, we examined the epistatic relationship between Stat3 and Tfcp2l1. We transfected *Stat3* null cells with expression vectors for *Klf4* or *Tfcp2l1*. After transfection, cells were cultured in serum-free medium containing PD and the selection agent hygromycin (Figure 5A). PB-Vector cells rapidly differentiated or died, as did PB-Klf4 cells and no stable transfectants were obtained (data not shown). In contrast, *Tfcp2l1* expressing cells expanded efficiently and could be maintained over several passages. They displayed undifferentiated morphology (Figure 5B) and were able to self-renew at clonal density (Figure 5C). We characterized the *Stat3* null cells expressing Tfcp2l1 by qRT–PCR and found that they expressed *Oct3/4*, *Sox2*, *Rex1*, and *Esrrb* at levels similar to wild-type ES cells in 2i. Reduced *Klf4* expression was observed, consistent with the absence of LIF/Stat3 signalling (Figure 5D). These data suggest that Tfcp2l1, but not Klf4, can replace Stat3 in ES cell self-renewal.

We then consulted the ES cell ChIP-seq compendium (http://bioinformatics.cscr.cam.ac.uk/ES_Cell_ChIP-seq_-compendium.html) to interrogate available ChIP-seq data sets (Martello *et al*, 2012). Unsupervised hierarchical clustering showed that Stat3 and Klf4 genome-wide binding profiles are closely related (Figure 5E). In contrast, Tfcp2l1 has a distinct binding profile and clusters with neither the 'core' factors nor Stat3 (Figure 5E). This is consistent with Tfcp2l1 acting downstream of Stat3 and additive to the 'core' factors to sustain ES cell self-renewal (Figure 5B).

### Tfcp2l1 mediates induction of pluripotency downstream of LIF/Stat3

LIF signalling through Stat3 plays a key role in induction of pluripotency. Activation of LIF/Stat3 is required for iPS cell formation (Tang *et al*, 2012; van Oosten *et al*, 2012), and is sufficient to reprogram post-implantation EpiSCs to naïve pluripotency (Yang *et al*, 2010). It is not known, however, whether the target genes that sustain self-renewal are the same as those that mediate reprogramming. To identify Stat3

**Figure 2** Identification of Stat3 primary targets in mouse ES cells. (**A**) Clonogenicity assay. Six hundred cells per well were plated either in 2i or in 2i + LIF on laminin-coated plates and stained for alkaline phosphatase (AP) after 5 days. Bars show the number of AP-positive colonies obtained. Mean and s.d. of three independent experiments is shown. (**B**) Top: Venn diagram showing overlap between genes upregulated (*P*-value<0.05) after 1 h of LIF treatment in *Stat3*$^{+/+}$ cells (orange), in *Stat3*$^{-/-}$ cells (green) and genes bound by Stat3 (see Materials and methods). Bottom: Venn diagram showing overlap between genes downregulated (*P*-value<0.05) after 1 h of LIF treatment in *Stat3*$^{+/+}$ cells (orange), in *Stat3*$^{-/-}$ cells (green) and genes bound by Stat3. (**C**) Flow chart illustrating the approach used to identify candidate genes that mediate self-renewal downstream of Stat3. See also Supplementary Table S1. (**D**) Absolute expression of the 13 genes identified as potential mediators of Stat3 activity in ES cells. Blue bars indicate expression in 2i and red bars indicate expression after 1 h of LIF treatment in 2i. (**E**) Gene expression analysis of ES cells cultured in 2i and treated with LIF for 1 h. Beta-actin served as an internal control. Mean and s.d. of two independent experiments is shown. (**F**) Gene expression analysis of ES cells cultured in 2i after LIF stimulation for 1 and 4 h in the presence of the protein-synthesis inhibitor Cycloheximide (CHX). Cells were expanded in 2i without LIF and treated with LIF for 1 or 4 h, in the presence of either CHX (50 µm/ml, blue bars) or DMSO (vehicle, orange bars). CHX or DMSO was added to the medium 30 min before LIF treatment, and maintained throughout the experiment, to ensure effective inhibition of protein synthesis. Beta-actin served as an internal control and data are normalized to unstimulated 2i cultures. Mean and s.d. of four biological replicates is shown. (**G**) Gene expression analysis of ES cells cultured under serum-free conditions in the presence of LIF and the Mek inhibitor PD0325901 (PD). LIF was withdrawn and the Jak inhibitor (1 µM) was applied for the indicated time. Beta-actin served as an internal control. Mean and s.d. of two independent experiments is shown.

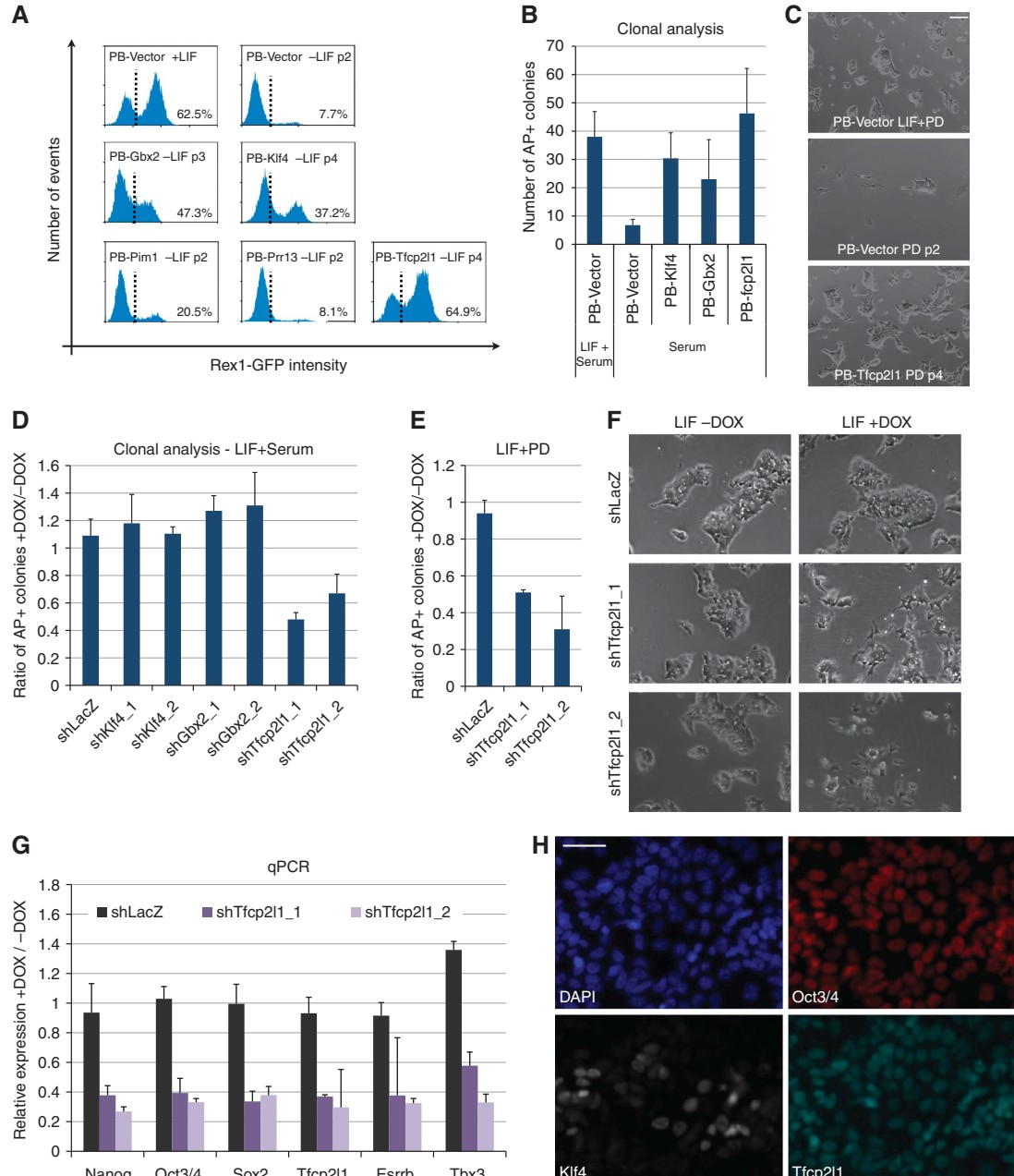

**Figure 3** Tfcp2l1 promotes ES cell self-renewal. (**A**) Flow cytometry analysis of Rex1-GFPd2 transfectants cultured in serum containing media either in the presence (+ LIF) or in the absence ( − LIF) of LIF. Rex1-GFPd2 cells were co-transfected with pBase helper plasmid and a piggyBac vector containing Gbx2, Klf4, Pim1, Prr13, and Tfcp2l1 or no cDNA (PB-Vector); transfected cells were selected for two passages with Hygromycin in LIF + serum. Only PB-Gbx2, PB-Klf4, and PB-Tfcp2l1 cells showed sustained self-renewal in the absence of LIF, expanding continuously for > 10 passages. The dashed line separates GFP-positive and GFP-negative cells and the percentage of GFP-positive cells is indicated. See also Supplementary Figure S1B. (**B**) Clonogenicity assay in serum. The indicated cell lines were plated at clonal density in serum containing media either in the presence (LIF + Serum) or in the absence (Serum) of LIF, and stained for alkaline phosphatase (AP) after 5 days. Bars show the number of AP + colonies obtained. Mean and s.d. of three independent experiments is shown. (**C**) Self-renewal in serum-free culture. PB-Vector and PB-Tfcp2l2 cells (described in Figure 3A) were cultured in the presence of LIF and the Mek inhibitor PD. After LIF withdrawal, PB-Vector cells rapidly differentiated or died (middle panel), whereas PB-Tfcp2l1 cells could be propagated with negligible differentiation for > 12 passages (bottom panel, see also Figure 4). (**D**) Clonogenicity assay on inducible knockdown ES cell lines. DOX-inducible shRNA constructs targeting the indicated genes were stably transfected in Rex1-GFPd2 cells (see Materials and methods). An shRNA targeting LacZ mRNA served as a negative control. Cells were plated at clonal density in LIF + serum media either in the presence or in the absence of DOX and stained for AP after 5 days. Bars show the ratio between colonies obtained in the presence and absence of DOX for the indicated shRNA lines. Mean and s.d. of two independent experiments is shown. (**E**) Clonogenicity assay on inducible knockdown ES cell lines under serum-free conditions. Cells expressing the indicated shRNA constructs were plated at clonal density in LIF + PD media either in the presence or in the absence of DOX and stained for AP after 5 days. Bars show the ratio between colonies obtained in the presence and absence of DOX for the indicated shRNA lines. Mean and s.d. of two independent experiments is shown. See also Supplementary Figure S1D. (**F**) Morphology of the indicated shRNA lines cultured in LIF either in the absence or in the presence of DOX for 4 days. Knockdown of *Tfcp2l1* resulted in differentiation. Scale bar, 100 μm. See also Supplementary Figure S1F. (**G**) Gene expression analysis of the indicated shRNA lines cultured in LIF + PD condition, either in the absence or in the presence of DOX for 4 days. Beta-actin served as an internal control and data are presented as ratios of + DOX and −DOX conditions. Mean and s.d. of two independent biological replicates is shown. (**H**) Fluorescence micrographs showing immunostaining for Oct3/4, Tfcp2l1, and Klf4 in wild-type ES cells cultured in LIF + serum. Scale bar, 100 μm.

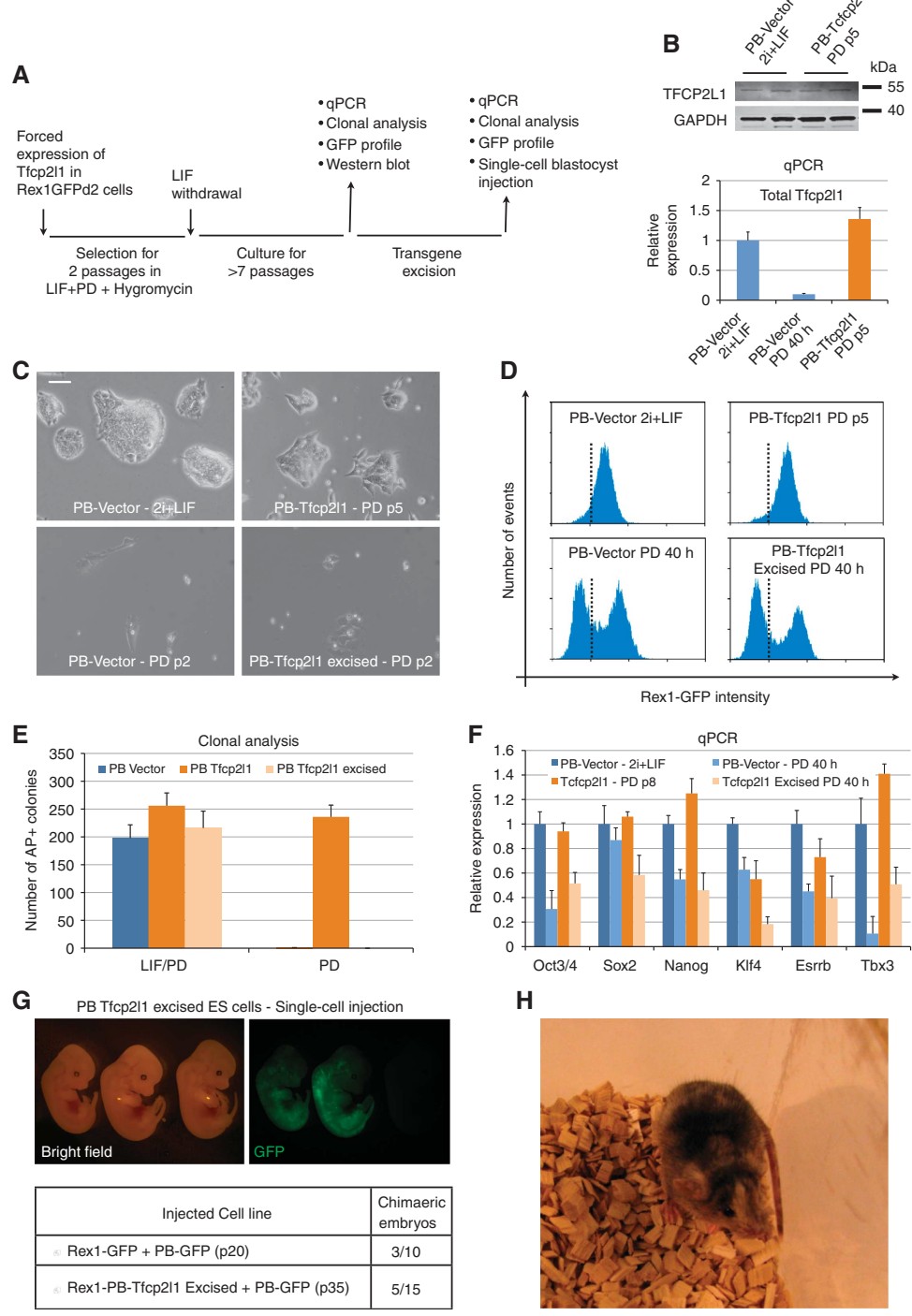

**Figure 4** Forced expression of Tfcp2l1 maintains pluripotency. (**A**) Experimental scheme for testing the sufficiency of Tfcp2l1 forced expression for the maintenance of ES cell pluripotency. (**B**) Top: western blot of Rex1GFPd2 cells transfected with an empty vector (PB-Vector) cultured in 2i + LIF, and PB-Tfcp2l1 cells cultured in PD for five passages. For each line, two biological replicates were loaded. GAPDH served as a loading control. Bottom: qRT-PCR analysis of total Tfcp2l1 expression of the indicated cell lines. Note that Tfcp2l1 is rapidly downregulated in PB-Vector cells cultured in PD alone. Beta-actin served as an internal control. (**C**) Representative pictures of PB-Vector and PB-Tfcp2l1 under the indicated culture conditions. Note that after excision of the transgene (PB-Tfcp2l1 excised) PB-Tfcp2l1 cells behave as PB-Vector cells. Scale bar, 100 μm. (**D**) Flow cytometry analysis of control (PB-Vector) and Tfcp2l1 expressing (PB-Tfcp2l1) cells under the indicated culture conditions. The dashed line separates GFP-positive and GFP-negative cells. (**E**) Clonogenicity assay of PB-Vector and PB-Tfcp2l1 cells before and after transgene excision. Note that PB-Tfcp2l1 cells behave as PB-Vector cells after excision of the Tfcp2l1 transgene (PB-Tfcp2l1 excised). Mean and s.d. of two independent biological replicates is shown. (**F**) Gene expression analysis of PB-Vector and PB-Tfcp2l1 cells cultured in the indicated conditions. Beta-actin was used as an endogenous control and data are normalized to PB-Vector cells cultured in 2i + LIF media. (**G**) Single-cell blastocyst injection. PB-Tfcp2l1 excised cells and Rex1GFPd2 cells were labelled with a constitutive GFP transgene (PB-GFP); single GFP-labelled cells were injected into blastocyst stage embryos; embryos were scored at mid-gestation (E12.5) for the presence of GFP-positive cells. Top: representative picture of chimaeric embryos derived after injection of single PB-Tfcp2l1 excised cells. The embryo on the right showed no chimaerism and serves as a control for GFP signal. Bottom: table summarizing the results obtained from two independent sections of injection. (**H**) Adult chimaeric animal showing widespread coat-colour contribution generated by single-cell blastocyst injection. Source data for this figure is available on the online supplementary information page.

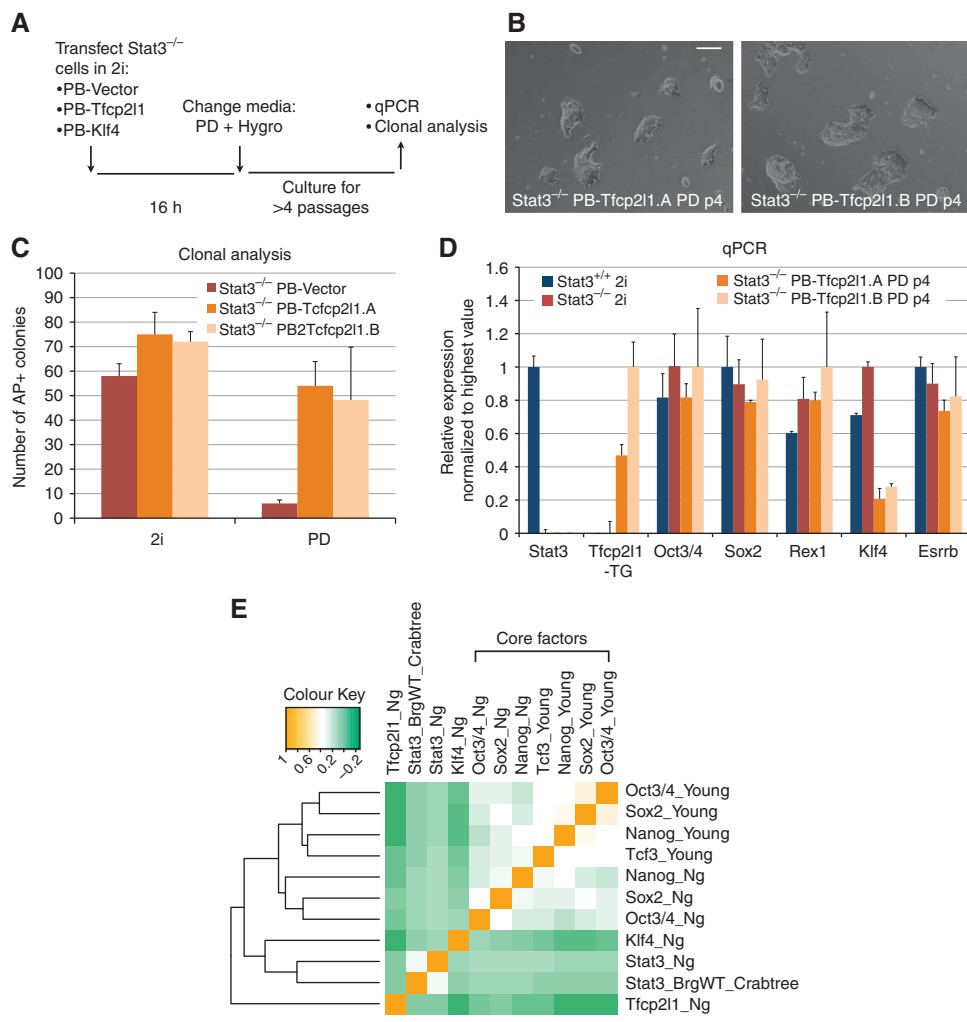

**Figure 5** Tfcp2l1 can replace Stat3 in ES cell self-renewal. (**A**) Experimental scheme for testing the ability of Tfcp2l1 and Klf4 to maintain self-renewal in *Stat3*$^{-/-}$ ES cells. (**B**) Representative pictures of PB-Tfcp2l1 transfected *Stat3*$^{-/-}$ ES cells cultured in PD for four passages. Two lines, generated in two independent experiments, are shown. Of note, neither PB-Vector nor PB-Klf4 transfected Stat3$^{-/-}$ cells could be recovered due to rapid differentiation and cell death. Scale bar, 100 µm. (**C**) Clonogenicity assay of PB-Vector and PB-Tfcp2l1 transfected *Stat3*$^{-/-}$ ES cells. The indicated cell lines were plated at clonal density either in 2i or in PD and stained for alkaline phosphatase (AP) after 5 days. *Stat3*$^{-/-}$ cells transfected with an empty vector (PB-Vector) were cultured in 2i before plating and serve as a control. Mean and s.d. of three independent experiments is shown. (**D**) Gene expression analysis of *Stat3*$^{-/-}$ cells expressing PB-Tfcp2l1 after prolonged culture in the presence of PD. Beta-actin was used as an endogenous control and data are normalized to the highest value. The Stat3 primers are designed to specifically recognize the wild-type allele of Stat3, confirming its absence in *Stat3*$^{-/-}$ lines. The Tfcp2l1-TG primers specifically recognize the exogenous Tfcp2l1 transcript. Mean and s.d. of three independent replicates is shown. (**E**) Hierarchical clustering of 11 genome-wide binding maps from the Mouse ES cell ChIP-seq Compendium. Hierarchical clustering and Pearson correlation coefficients were used to display all pairwise comparisons in a clustered heatmap. Colours in the heatmap show the level of correlation for all pairwise comparisons (orange = 1; white = 0.5, green = 0). Factors have been clustered along both axes according to the level of correlation. Note that, as previously reported, the 'core' factors Oct3/4, Sox2, and Nanog cluster together; Klf4 clusters together with Stat3 whereas Tfcp2l1 has a distinct binding profile.

targets active during reprogramming, we used embryo-derived EpiSCs that stably express a chimaeric gp130 receptor, GY118F, which elicits hyperactivation of endogenous Stat3 in response to granulocyte colony stimulating factor (GCSF) (Burdon *et al*, 1999; Yang *et al*, 2010). Exposure of these EpiSCs to GCSF for 48 h in 2i is sufficient to induce formation of a modest number of reprogrammed colonies (Figure 6B, see also Yang *et al*, 2010). Importantly, exposing EpiSCs to LIF is not effective, suggesting that activation of Stat3 is limiting for induction of key target gene(s) to a level sufficient to drive reprogramming. We examined expression of a panel of candidate genes in EpiSCs exposed to either 2i + LIF or

2i + GCSF for 48 h in comparison with EpiSCs cultured in FGF2 and Activin A (F/A). We also included ES cells cultured in 2i + LIF as a reference. We chose 30 Stat3 targets, based on the present and previous studies (Bourillot *et al*, 2009), and determined their expression by qRT–PCR. We searched for genes that were induced in EpiSCs and, more specifically, were upregulated to higher levels in 2i/GCSF compared to 2i + LIF (Figure 6C, compare the second to the first column in the heatmap). Interestingly, we found that *Klf4* and *Myc*, two reprogramming factors previously reported as Stat3 targets, were not induced by GCSF in EpiSCs, whereas four transcription factors, *Klf5*, *Gbx2*, *Pim1*, and *Tfcp2l1*, were upregulated to levels comparable to, or higher than, their expression in ES

cells (Figure 6C, blue box in the heatmap and histograms). We tested whether induction of these four Stat3 targets was required for reprogramming by knockdown. We performed siRNA transfection followed by GCSF stimulation and scored the number of Oct4-GFP-positive colonies after 4 days in 2i + LIF (Figure 6D, top). *Stat3* knockdown resulted in a ~10-fold reduction, confirming that Stat3 mediates induction of pluripotency downstream of the chimaeric GCFS/gp130 receptor. We found that knockdown of *Gbx2* and *Klf5* had no effect, and knockdown of *Pim1* resulted in a ~2-fold decrease in colony number. However, *Tfcp2l1* ablation substantially reduced the number of colonies recovered. This suggests that *Tfcp2l1* plays a critical role in induction of naïve pluripotency downstream of Stat3.

We then tested whether *Tfcp2l1* expression is sufficient to convert EpiSCs into naïve pluripotency. We found that Oct4-GFP reporter EpiSCs (O4GIP) stably transfected with a *Tfcp2l1* expression vector efficiently converted into naïve pluripotency when exposed to 2i (Figure 6E). Importantly, addition of LIF resulted in only a modest further increase in the number of colonies, suggesting that forced expression of *Tfcp2l1* could largely recapitulate the contribution of LIF to EpiSC reprogramming. The reprogrammed colonies obtained could be passaged in 2i + LIF and gave rise to stable iPS cell lines (Figure 6F). Gene expression analysis confirmed activation of the naïve markers *Tbx3* and *Esrrb* and downregulation of the EpiSC markers *Fgf5* and *Sox17* (Figure 6G).

EpiSCs are variable and one line (GOF18) derived on feeders in the presence of FGF2 has been reported to convert spontaneously to naïve pluripotency when exposed to 2i + LIF (Han *et al*, 2010). We found that GOF18 cells retained this ability even after prolonged culture under feeder-free conditions in the presence of FGF2 and Activin A. The conversion frequency in 2i alone was extremely low, but was greatly increased by addition of LIF (Supplementary Figure S2A). Moreover, we found that addition of LIF for only the first 48 h gave rise to the same number of colonies as continuous exposure (Supplementary Figure S2A, compare second and third bar). Therefore, we measured the expression of Stat3 targets by qPCR in GOF18 cells exposed to either 2i or 2i + LIF for 48 h. Both *Gbx2* and *Tfcp2l1* were expressed in these EpiSCs in 2i, but significantly only Tfcp2l1 was further upregulated by LIF (Supplementary Figure S2B). We transiently transfected GOF18 cells with a *Tfcp2l1* expression vector and 24 h after transfection replated them in 2i without LIF. After 6 days, we scored the number of Oct4-GFP-positive colonies. We observed an ~10-fold increase in colony formation after *Tfcp2l1* transfection (Supplementary Figure S2C, left), indicating that a transient increase in *Tfcp2l1* expression is sufficient to drive the reprogramming process. The iPS cells generated after *Tfcp2l1* expression could be expanded in 2i + LIF over multiple passages, showing undifferentiated morphology, stable expression of the Oct4-GFP reporter (Supplementary Figure S2C, right panels) and of naïve pluripotency markers, and a concomitant reduction in EpiSC markers (Supplementary Figure S2D).

We analysed data on a panel of EpiSC lines (Bernemann *et al*, 2011) and found no significant differences in expression of LIF/Stat3 pathway components between lines that convert spontaneously and lines that do not (Supplementary Figure S2E, compare red and blue bars). We also found that both O4GIP and GOF18 cells express *Tfcp2l1* mRNA at similar very low levels (Supplementary Figure S2F). Moreover, Tfcp2l1 protein 1 could not be detected by immunofluorescence in GOF18 cells, excluding the possibility of a rare sub-population of expressing cells (Supplementary Figure S2G). We conclude that spontaneous conversion into naïve pluripotency does not correlate with increased *Tfcp2l1* expression in the EpiSC state, but rather with the ability to activate *Tfcp2l1*.

Finally, to confirm functional naïve pluripotent status after Tfcp2l1-mediated reprogramming, we transfected O4GIP-derived iPS cells with a constitutive GFP expression vector, excised the *Tfcp2l1* transgene by *Cre* plasmid transfection, and carried out blastocyst injection. Two out of six transferred embryos showed widespread GFP expression at mid-gestation (Figure 6H). Taken together, these results suggest that Tfcp2l1 plays a major role downstream of LIF/Stat3 in the conversion of EpiSCs into authentic naïve pluripotency.

## Discussion

The LIF/Stat3 pathway plays crucial facultative roles in mouse ES cell self-renewal (Smith *et al*, 1988; Williams *et al*, 1988; Niwa *et al*, 1998), maintenance of pluripotent epiblast during embryonic diapause (Nichols *et al*, 2001), generation of EG cells (Leitch *et al*, 2013), and molecular induction of pluripotency (Yang *et al*, 2010; Tang *et al*, 2012; van Oosten *et al*, 2012). How LIF input is integrated with the pluripotency gene regulatory network has remained unclear, however (Niwa *et al*, 1998, 2009; Nichols and Smith, 2012). cMyc has been proposed as a mediator (Cartwright *et al*, 2005), but neither *cMyc* nor *NMyc* are appreciably induced by LIF in ES cells (Hall *et al*, 2009). *Klf4* has been shown to be a direct target of Stat3 (Li *et al*, 2005; Hall *et al*, 2009; Niwa *et al*, 2009), but is neither required nor sufficient to account for the potent effects of LIF stimulation. A significant role for alternative modes of LIF signalling has also been invoked, either through PI3 kinase (Welham *et al*, 2007; Niwa *et al*, 2009) or by JAK modification of chromatin (Griffiths *et al*, 2011).

To resolve this issue, we first confirmed that Stat3 null cells have functional ES cell identity and potency yet show no self-renewal response to LIF. We then sought the missing target(s) of Stat3 and identified the transcription factor Tfcp2l1. Tfcp2l1, also known as Crtr1, has previously been described in ES cells (Pelton *et al*, 2002; Ivanova *et al*, 2006; Chen *et al*, 2008), but its relationship to LIF/Stat3 has gone unnoticed. *Tfcp2l1* is regulated directly by activated Stat3 and induced to high levels. Gain- and loss-of-function perturbations established that Tfcp2l1 is both necessary and largely sufficient to mediate the effects of LIF on ES cell self-renewal in either serum or serum-free culture conditions, in contrast to previously identified targets, *Klf4*, *Gbx2*, and *Pim1*. Strikingly, forcing expression of *Tfcp2l1* at close to endogenous levels was sufficient to phenocopy LIF stimulation. Finally, we showed that Tcfcp2l1 is epistatic to Stat3 in sustaining ES cell self-renewal (Figure 5). From these results, we conclude that Tfcp2l1 is the major mediator of ES cell self-renewal downstream of LIF/Stat3, and that it exerts functions that cannot be compensated by other LIF targets. We previously noted that *Tfcp2l1* is also induced by GSK3 inhibition (Martello *et al*, 2012). It may therefore constitute the postulated point of intersection between these two mechanisms for supporting ES cell self-renewal (Hao *et al*,

2006; Ogawa *et al*, 2006; Wray *et al*, 2010). Notably, however, while Tfcp2l1 appears sufficient to replace LIF/Stat3, it cannot reproduce the full effect of GSK3 inhibition (Figure 3A in Martello *et al*, 2012).

We also investigated mediators of LIF/Stat3 action during reprogramming. Klf4 is a canonical reprogramming factor (Takahashi and Yamanaka, 2006). However, LIF is required for efficient reprogramming of both somatic cells and EpiSCs

even when *Klf4* is overexpressed (Yang *et al*, 2010; Tang *et al*, 2012), suggesting a requirement for other LIF/Stat3 target(s). Analysis of the initial phase of EpiSC reprogramming did not show induction of known reprogramming factors such as *Klf4* and *cMyc*, consistent with previous observations (Yang *et al*, 2010). In contrast, *Gbx2, Klf5, Pim1,* and *Tfcp2l1* were induced (Figure 6C). Transient expression of *Tfcp2l1* was sufficient to convert EpiSCs into naïve pluripotency without

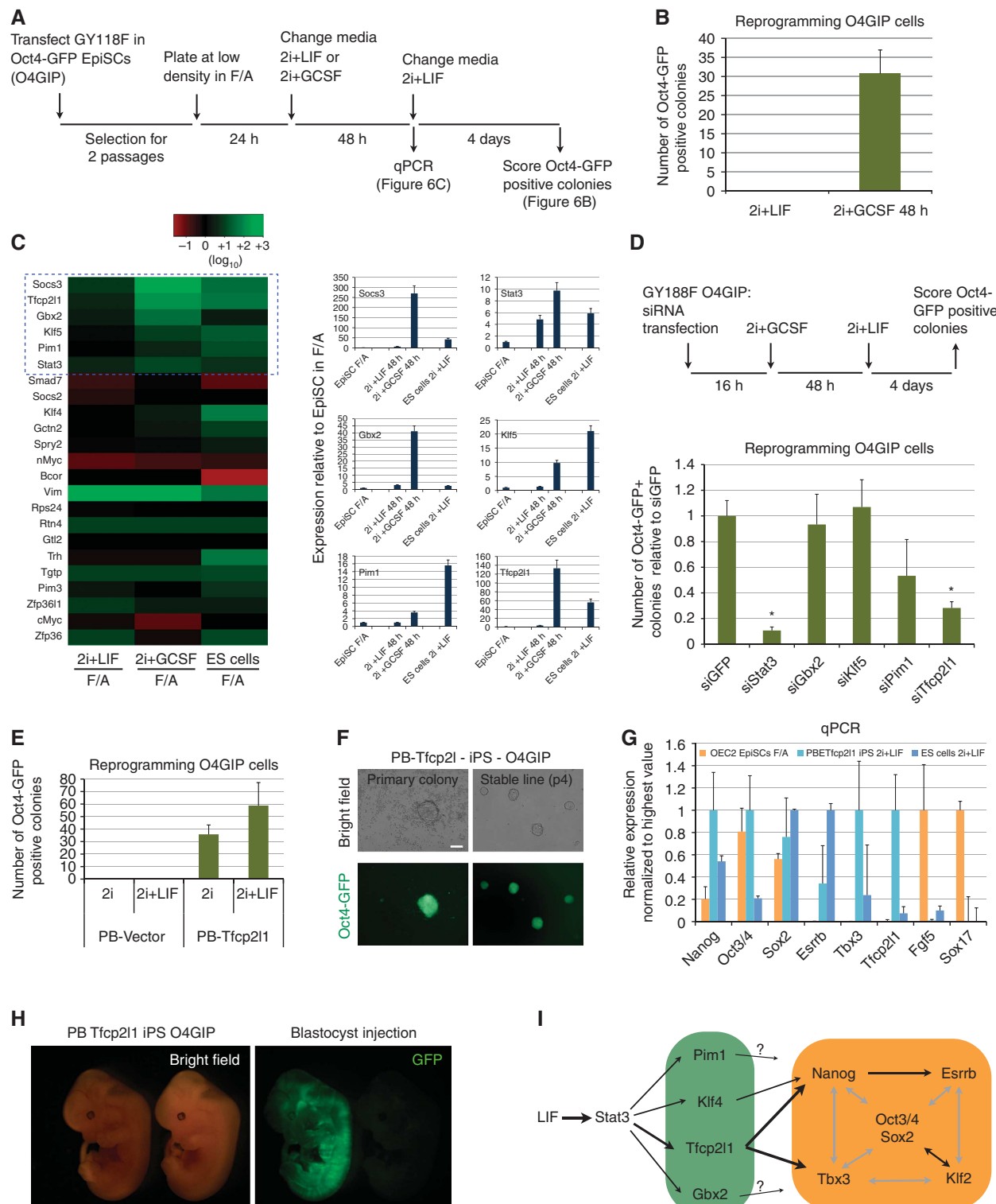

LIF stimulation, while *Tfcp2l1* knockdown caused a dramatic reduction in reprogramming efficiency (Figure 6D). Importantly, when *Tfcp2l1* is knocked down in ES cells in 2i + LIF there was no significant compromise to self-renewal (Supplementary Figure S1D and E). This suggests that Tfcp2l1 has a specific role in resetting of naive pluripotency in addition to its action in self-renewal.

Tfcp2l1 is a member of the CP2 family of transcription factors, which are generally considered to be activators of transcription (Lim *et al*, 1993; Bing *et al*, 1999). This has been demonstrated for Tcfp2l1 in ES cells (To *et al*, 2010), although in some contexts it may act as a repressor (Rodda *et al*, 2002). Interestingly, genome location data indicate that *Tbx3*, *Esrrb*, and *Nanog* are bound by Tfcp2l1 (data from Chen *et al*, 2008 and ES cell ChIP-seq Compendium) and upon *Tfcp2l1* knockdown we noted downregulation of the three genes (Figure 3G). Similar effects of *Tfcp2l1* depletion are also apparent in data from a recent microarray study (Nishiyama *et al*, 2013; Supplementary Figure S3A and B). Furthermore, we observed that upon overexpression of *Tfcp2l1*, *Tbx3* and Nanog are significantly upregulated (Figure 6G; Supplementary Figures S2D, S3A and B and data not shown). These observations suggest that Tfcp2l1 sustains ES cell self-renewal in large part through transcriptional activation of *Nanog* and *Tbx3*, both of which have been proposed to lie genetically downstream of LIF (Niwa *et al*, 2009) but neither of which are direct targets (Bourillot *et al*, 2009; Hall *et al*, 2009).

*Tfcp2l1* is expressed in the inner cell mass of the mouse blastocyst and is downregulated shortly after implantation (Pelton *et al*, 2002; Guo *et al*, 2010). It is rapidly downregulated when ES cells start to differentiate (Ivanova *et al*, 2006) and is absent from mouse EpiSCs (Figure 6) in common with other naïve pluripotency markers such as *Esrrb*, *Rex1*, and *Klf2/4* (Tesar *et al*, 2007; Nichols and Smith, 2012). Human pluripotent ES cells are considered to be in a primed pluripotent state closer to mouse EpiSCs than ES cells. *Tfcp2l1* is detected in the human ICM, but downregulated during derivation of human ES cells (O'Leary *et al*, 2012). Human ES cells have been reported to convert into a naïve-like state by overexpression of *Klf2*, *Klf4*, and *Oct3/4* (Hanna *et al*, 2010). Interestingly, inspection of the microarray data from that study reveals that Tfcp2l1 is highly upregulated. It is possible therefore that Tfcp2l1 may play a role in generating and stabilizing a human naïve pluripotent state.

LIF is a stimulus that confers self-renewal and can promote emergence of the naïve pluripotent gene regulatory network, but is not itself integral to pluripotent identity (Nichols *et al*, 2001; Smith, 2001; Ying *et al*, 2008; Nichols and Smith, 2012). Consistent with this, Tfcp2l1 genome occupancy does not cluster with that of canonical pluripotency factors Oct4, Sox2, and Nanog. Instead, it binds to a largely distinct, and broad, set of genes. Significantly, however, these targets include key naïve pluripotency factors *Nanog* and *Tbx3* previously shown to support ES cell self-renewal (Chambers *et al*, 2003; Ivanova *et al*, 2006; Niwa *et al*, 2009). Furthermore, the *Tfcp2l1* promoter is bound by Oct4, Sox2, Nanog, Esrrb, and Klf4 (ES cell ChIP-seq compendium), and is targeted by the pluripotency repressor Tcf3 (Martello *et al*, 2012), indicating that Tfcp2l1 is hard-wired into the pluripotency gene regulatory network. We conclude that Tfcp2l1 is a previously overlooked transcriptional regulator that plays a pivotal role in the capability to generate, derive, and propagate naïve pluripotent stem cells. Although mutation of *Tfcp2l1* is reported to be compatible with mouse embryo development, the characterized gene trap mutation may not create a null allele since residual full-length transcript is detectable (Yamaguchi *et al*, 2005). It will therefore be of interest to determine the effects of complete deletion of *Tfcp2l1* on the emergence of pluripotency in the blastocyst, and epiblast maintenance during diapause when the Stat3 pathway is indispensable (Nichols *et al*, 2001).

# Materials and methods

### RNA-seq library construction
RNA was extracted using the TRIzol method (Invitrogen) followed by treatment with TURBO DNase (Ambion). Ribosomal RNA was depleted using RiboMinus (Invitrogen), and the remaining RNA was sheared by ultrasonication on a Covaris S2 for 90 s with the

**Figure 6** Tfcp2l1 mediates resetting of naive pluripotency. (**A**) Experimental scheme used to identify genes activated by the chimaeric GCSF/ LIF-R receptor (GY118F) during EpiSC reprogramming. (**B**) Number of Oct4-GFP colonies obtained from O4GIP cells 6 days after induction of reprogramming. Note that activation for 48 h of the chimaeric GCSF/LIF-R receptor is sufficient to convert EpiSCs efficiently to ground-state pluripotency. (**C**) Left: heatmap showing the relative expression of the indicated genes in EpiSCs exposed to either 2i + LIF for 48 h (first column) or 2i + GCSF (second column); as a reference the expression in mouse ES cells is shown (third column). The fold change expression relative to EpiSCs cultured in bFGF and Activin A condition (F/A) is shown and beta-actin serves as an internal control. Right: histograms showing the relative expression of selected Stat3 targets. Stat3 itself and Socs3 served as positive controls, whereas Gbx2, Klf5, Pim1, and Tfcp2l1 were chosen for further functional validations. (**D**) Top: experimental scheme for testing the role of the indicated genes in the process of reprogramming. O4GIP EpiSCs expressing the chimaeric GCSF/LIF-R receptor were transfected with the indicated siRNAs in bFGF and Activin A media; after 16 h, they were exposed to GCSF in 2i media for 48 h, followed by 4 days of culture in 2i + LIF media. Bottom: Histogram showing the number of Oct4-GFP-positive colonies obtained relative to siGFP-transfected cells. Mean and s.d. of three independent experiments is shown. *P<0.01 (t-test) compared to siGFP control. (**E**) O4GIP EpiSCs were transfected with PB-Tfcp2l1 and an empty piggyBac vector control (PB-Vector) in bFGF and Activin A media; after 3 days of Hygro selection, transfectants were replated at low density and cultured in either 2i or 2i + LIF media for 6 days; the number of Oct4-GFP-positive colonies is shown as mean and s.d. of three independent experiments. Similar results were obtained by transient transfection of PB-Tfcp2l1 with no selection. (**F**) Representative pictures of primary Oct4-GFP colonies emerged after 5 days of reprogramming (left) and of a stable iPS line generated after PB-Tfcp2l1 transfection. Scale bar, 100 μm. (**G**) Gene expression analysis of parental O4GIP EpiSCs (orange), PB-Tfcp2l1 iPS cells (light blue), and wild-type ES cells (blue). Beta-actin was used as an endogenous control and data are normalized to the highest value. Mean and s.d. of two independent biological replicates is shown. (**H**) O4GIP iPS cells were transfected with a constitutive GFP expression plasmid and injected into blastocyst stage embryos after excision of the Tfcp2l1 transgene; embryos were scored at E12.5 for the presence of GFP-positive cells and two embryos out of six showed widespread chimaerism. One of these chimaeras is shown. The embryo on the right showed no chimaerism and served as a control for GFP signal. (**I**) Schematic diagram of LIF/Stat3 input to the pluripotency network. Among several transcription factor mediators Tfcp2l1 is of paramount importance and is uniquely required to promote self-renewal through integration into the core pluripotency gene regulatory network. Grey lines represent generic interconnectivity between core factors.

following parameter settings: Duty Cycle = 10, Cycles per Burst = 200, Intensity = 5. Fragmented RNA was reverse-transcribed with SuperScript III (Invitrogen) at 50°C for 2 h using random hexamer and oligo-dT primers (10:1) in the presence of 6 μg/ml actinomycin D to inhibit the generation of second-strand products. Second-strand cDNA was synthesized by DNA Polymerase I for 2 h at 16°C in the presence of RNase H and with dUTPs substituted for dTTPs. End repair of double-strand cDNAs was carried out with T4 DNA polymerase and T4 polynucleotide kinase (New England Biolabs). Blunt-end, 3′-phosphorylated products were 3′-adenylated by exo-Klenow fragment in the presence of dATPs and ligated to sequencing adapters (Illumina) by T4 DNA ligase (New England Biolabs) at 20°C for 30 min. Following adapter ligation, the second strand of the library constructs was digested with uracil DNA glycosylase (UDG) and apurinic/apyrimidinic endonuclease 1 (APE 1) for 30 min at 37°C. PCR amplification of first-strand library constructs was carried out with Phusion DNA polymerase (Finnzymes) for 15 cycles. Purification of reaction products between each step was performed with Ampure XP paramagnetic beads (Beckman Coulter). The molarity and size distribution of the libraries was assessed by DNA 1000 microfluidic chips on the Agilent 2100 Bioanalyzer. Sequencing was performed on the Illumina GAIIx yielding 35–40 M 105 bp reads per sample. RNA-seq data are available in the ArrayExpress repository under accession E-MTAB-1796.

### RNA-seq data analysis

Reads were aligned to the mouse reference genome (build mm10) using GSNAP version of 2012-05-24 (Wu and Nacu, 2010). The aligner was provided with known splice sites based on Ensembl version 68 (Flicek et al, 2012), but enabled to identify novel splice sites. A maximum of 10 mismatches were allowed for read alignment. Gene counts were calculated using the htseq-count utility (version 0.5.3p3, http://www-huber.embl.de/users/anders/HTSeq/doc/count.html) and used as an input for differential gene expression analysis with DESeq version 1.10.1 (Anders and Huber, 2010). Genes with a $P$-value of $<0.05$ (Benjamini–Hochberg adjusted) were selected for further analysis.

### ChIP-seq data analysis

We obtained ChIP-seq data for Stat3 and a GFP control from a previous study (Chen et al, 2008). Reads were mapped to the mouse genome (build mm10) using bowtie version 0.12.8 (Langmead et al, 2009), discarding non-unique alignments. Peak detection was performed using MACS version 1.4.1 (Zhang et al, 2008). Each peak was associated with up to two genes as follows: the most proximal gene on each strand was identified by examining 50 kb upstream and downstream of the peak. Peaks were assigned to intersect genes unless an overlapping gene was found on the opposite strand. For the heatmap in Figure 5E a binary peak matrix was generated as described (Martello et al, 2012), analysed by unsupervised hierarchical clustering using Pearson's correlation coefficients, and displayed using the heatmap function in R. The ChIP-seq data used for target gene intersection in Figure 5F are Esrrb—GSE11431 (GSM288355) and TFCP2L1—GSE11431. All raw and processed data used in this study are available at http://bioinformatics.cscr.cam.ac.uk/ES_Cell_ChIP-seq_compendium.html

### ES cell culture

ES cells were cultured without feeders on plastic coated with 0.1% gelatine (Sigma, cat. G1890) and replated every 3 days at a split ratio of 1 in 10 following dissociation with Accutase (PAA, cat. L11-007). Cells were cultured either in the GMEM (Sigma, cat. G5154) supplemented with 10% FCS (Sigma, cat. F7524), 100 mM 2-mercaptoethanol (Sigma, cat. M7522), 1 × MEM non-essential amino acids (Invitrogen, cat. 1140-036), 2 mM L-glutamine, 1 mM sodium pyruvate (both from Invitrogen), and 100 units/ml LIF, or in the serum-free media N2B27 (NDiff N2B27 base medium, Stem Cell Sciences Ltd, cat. SCS-SF-NB-02) supplemented, as indicated, with small-molecule inhibitors PD (1 μM, PD0325901) and CH (3 μM, CHIR99021) and LIF prepared in-house. Colony forming assays were carried out by plating 60 ES cells/cm² on plates coated with laminin (Sigma, cat. L2020). Plates were fixed and stained for alkaline phosphatase (Sigma, cat. 86 R-1KT) according to the manufacturer's protocol. Plates were scanned using a CellCelector (Aviso) and

scored manually. EpiSCs were cultured as previously described (Guo et al, 2009). O4GIP (OEC-2 line) was described in Betschinger et al (2013). GOF18 was described in (Han et al (2010). For DNA transfection, we used Lipofectamine 2000 (Invitrogen, cat. 11668030) and performed reverse transfection. For one well of a 6-well plate (10 cm²), we used 6 μl of transfection reagent, 2 μg of plasmid DNA, and 300 000 cells in 2 ml of N2B27 medium. The medium was changed after overnight incubation. The piggyBac vectors used for overexpression followed by transgene excision have been described in Guo et al (2009). The transgene is flanked by loxP sites, allowing efficient excision by transfection with Cre.

### Gene expression analysis by quantitative RT–PCR with reverse transcription

Total RNA was isolated using the RNeasy kit (Qiagen) and 500 ng used for cDNA synthesis using SuperScript III (Invitrogen) and oligo-dT primers. Quantitative real-time PCR was carried out with SYBR green detection. Primers are detailed in Supplementary Table S2. Technical replicates were carried out for all reactions.

### RNAi experiments

siRNAs were transfected at a final concentration of 40 nM using Dharmafect 1 (Dharmacon, cat. T-2001-01), following the protocol for reverse transfection. For a 12-well plate (4 cm²), we used 2 μl of transfection reagent, 2 μl of 20 μM siRNA solution, and 30 000 ES cells in 1 ml of N2B27 medium. The medium was changed after overnight incubation. siRNAs were purchased from Qiagen (Flexitube GeneSolution, see Supplementary Table S3).

### Inducible shRNA

To generate a Dox-inducible shRNA system, we inserted into a piggyBac vector the murine pre-miR-155 hairpin (from the BLOCK-IT Pol2 miR RNAi system; Invitrogen) downstream of a third-generation Tet-responsive promoter (TRE3G). Sequences targeting the gene of interest (Supplementary Table S4) were cloned into the miR-155 hairpin by homologous recombination using the Infusion HD cloning kit (Clontech). The third-generation TET activator (TET3G) was cloned in a piggyBac vector under control of a constitutive CAG promoter. Cells were co-transfected with pBase helper plasmid and the two piggyBac vectors containing both the TRE3G-shRNA and the TET3G, and cultured in the presence of selection agent for 7 days prior to experimental assays.

### Flow cytometry

After treatment with Accutase, live ES cells were resuspended in PBS with 3% FCS and 0.05 nM ToPro-3 (Invitrogen) was added at a concentration of 0.05 nM to detect dead cells. Flow cytometry analyses were performed using a Dako Cytomation CyAn ADP high-performance cytometer with Summit software.

### Immunostaining

Cells were fixed for 10 min in 4% PFA at Room Temperature (RT), permeabilized for 5 min in PBS + 0.2% Triton X at RT, and blocked for 30 min in PBS + 3% donkey serum at RT. Cells were incubated overnight at 4°C with the primary antibodies (anti-Tcfcp2l1: rabbit IgG, Abcam 123354, 1:400; anti-Klf4: goat IgG, R&D Systems AF3158, 1:400; anti-Oct4: mouse IgG, Santa Cruz C-10, used at a 1:300 dilution). After washing in PBS, the cells were incubated with Alexa Fluor secondary antibodies (Invitrogen, 1:500 in PBS + 3% serum), for 30 min at RT. After DAPI staining, images were acquired using a Zeiss AxioObserver D1 microscope. Automated single-cell image quantification was performed using CellProfile (Broad Institute—Carpenter et al, 2006).

### Immunoblotting

Immunoblotting was performed as previously described in Yang et al (2010). Antibodies used are rabbit polyclonal anti-Tfcp2l1 (Abcam, AB123354, 1:500 dilution in 1% milk) and mouse monoclonal anti-GAPDH (Sigma-Aldrich, G8795, 1:1000 in 5% milk).

### Supplementary data

Supplementary data are available at The EMBO Journal Online (http://www.embojournal.org).

## Acknowledgements

We thank Tüzer Kalkan, Jörg Betschinger, Ge Guo and Alison McGarvey for advice, reagents and help with experiments; Hans Schöler for providing GOF18 EpiSCs; William Mansfield, Charles-Etienne Dumeau, Peter Humphreys, and Andy Riddell for specialist technical support. We also thank the EMBL Genomics Core Facility for sequencing and Tamara Steijger for analysis of RNA-seq and ChIP-seq data. We are grateful to Qi Long Ying for sharing information prior to publication. This study was funded by the Biotechnology and Biological Sciences Research Council of the United Kingdom and the Swiss National Science Foundation Sinergia Programme. GM is recipient of a Human Frontier Science Program fellowship and AS is a Medical Research Council Professor.

*Author contributions*: GM and AS conceived the study and wrote the manuscript. GM performed and analysed the experiments. PB generated RNA-seq libraries and oversaw analysis.

## Conflict of interest

The authors declare that they have no conflict of interest.

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
