## [Review Process File · The EMBO Journal]

Manuscript EMBO-2013-85518

Identification of the Missing Pluripotency Mediator Downstream of Leukaemia Inhibitory Factor

Graziano Martello, Paul Bertone and Austin Smith

*Corresponding authors: Austin Smith and Graziano Martello, Wellcome Trust - MRC Cambridge
Stem Cell Institute*

Review timeline:

Submission date:	30 April 2013
Editorial Decision:	22 May 2013
Revision received:	11 July 2013
Accepted	16 July 2013

Transaction Report:

Editor: Thomas Schwarz-Romond

1st Editorial Decision

22 May 2013

Thank you very much for submitting your study characterizing Tfcp2l1 as critical pluripotency factor downstream of LIF for consideration to The EMBO Journal editorial office.

From the attached reports you will easily recognize that all three scientists appreciate novelty and general interest, hence suitability of your findings for The EMBO Journal. Despite this, refs#2 and #3 suggest a few further reaching experiments to offer initial insight(s) into Tfcp2l1's functional contribution in this context (major points 1 and 2 ref#2). Further analyses should aim to clarify Tfcp2l1's epistatic position within the pluripotency network, particularly with regard to Esrrb in maintenance/establishment of naïve pluripotency.

Rather certain that you are in a strong position to develop the manuscript along these constructive comments in a timely manner, I am delighted to invite submission of an appropriately revised manuscript.

Please be reminded that The EMBO Journal considers only one round of revisions and the ultimate decision on publication will dependent on the outline and strength of your revised manuscript.

I am very much looking forward to your amended study and remain with best regards.

REFEREE REPORTS:

Referee #1:

The paper by Martello et al, "Identification of the critical pluripotency factor downstream of LIF" describes the identification of Tfc211 as a target of Stat3 regulation and a key mediator of LIF signaling that maintains ES cell pluripotency.

This is a very nice study. The authors logically analyzed transcriptome data to identify plausible candidates of LIF/Stat3 signaling. They proceeded to perform the necessary experiments, perturbing Tfc211 expression upward and downward, to establish the role of this transcription factor. The paper is well written, the experiments were well executed and the data interpretation was sound. It was a pleasure to review this paper.

This reviewer only offers a few minor comments that would improve the paper.

1. The title is a bit grandiose in the stating "the critical pluripotency factor." At least say "a critical pluripotency factor." Perhaps even better to simply state that Tfc211 is the factor.
2. The data figures are rather packed. Perhaps some information could be moved to supplementary section. For example, Figure 1 mostly reiterates what was already published in Ying et al 2008, so might be better as supplementary data.
3. The 6136 genes expressed in ES cells and bound by Tfc211 should be included as supplementary data.
4. In results section, it is not clear how it is concluded that "This widespread binding suggests that Tfc211 may be a rather general regulator..." Please clarify, move to discussion, or remove.
5. Legend in Figure 3 has a typo: written as "3E", but should be "3H".
6. Scale bars should be added to figures 3H and 4C.
7. Pim1 comes up several times in the results, but gets no mention in the discussion. Perhaps some comment is warranted.
8. Be good to comment about whether other pluripotency TF's bind at Tfc211.
9. Please provide section headings in results section.

Referee #2:

In this manuscript, Graziano Martello and colleagues uncover Tfc211 as a critical downstream target of LIF/Stat3 signalling pathway in mouse ES cells. Here, the authors take advantage of Stat3^{-/-} ES cells, which retain their self-renewal and pluripotent abilities if maintained in the presence of Mek and Gsk3 (2i) inhibitors. Other conditions (serum+LIF and PD+LIF) in contrast severely compromise Stat3^{-/-} ES cell self-renewal, as the cells fail to respond to LIF signal transduction. Combining RNA-Seq analysis in LIF-stimulated Stat3^{+/+} and Stat3^{-/-} ES cells and Stat3 ChIP-Seq dataset, they identify and validate Tfc211 as a primary LIF/Stat3 target. Accordingly, Tfc211 forced expression can substitute for LIF/Stat3 in Stat3^{-/-} ES cells and furthermore supports an undifferentiated ES cell state in non-permissive conditions. Conversely, conditional loss of Tfc211 is detrimental to ES cells unless grown in 2i+LIF conditions. Interrogation of Tfc211 ChIP-Seq dataset indicates that Tfc211 might partly act 'outside' the core transcriptional network to sustain ES cell self-renewal, with many Tfc211-bound target genes being expressed in 2i+LIF conditions. Beyond its role in maintaining the ES cell identity, Tfc211 is also revealed as a potent mediator of EpiS cell reprogramming to naïve pluripotency. Overall, I believe that this manuscript would be a

nice candidate for EMBO and would be of interest to a broad readership in the fields of transcription, pluripotency and reprogramming.

Major concerns:

1- The study indicates that Tcfp2l1 potentially functions in parallel to the core transcriptional network to sustain ES cell self-renewal but falls short in explaining how Tcfp2l1 operates in this context. Figure 3F-G suggests that Tcfp2l1 plays a role in ES cell survival in addition to restraining differentiation. Could the authors clarify whether Tcfp2l1 conditional deletion indeed leads to an increased incidence of apoptosis (potentially rescued in the presence of caspase inhibitors) and whether "surviving" colonies retain a positive AP-staining? Important clues might also come from detailed gene ontology analysis of Tcfp2l1-bound targets (Figure 5E) with a specific focus on Tcfp2l1-bound genes that are expressed in 2i+LIF conditions as compared to serum + LIF (Marks et al. 2012).

2- The authors report that Tcfp2l1 is homogeneously expressed in ES cells in contrast to Klf4 and many other pluripotency transcription factors (Figure 3H), suggesting that Tcfp2l1 "... may underlie the capacity of LIF to sustain self-renewal efficiently in serum when other factors are more restricted". This is an interesting observation that needs to be further explored, as this could provide additional insights into how Tcfp2l1 operates in ES cells. The authors should perhaps take advantage of the recently established Esrrb -/- ES cells (Martello et al. 2012), as these cells were shown to strictly rely on LIF to accommodate the loss of Esrrb.

3- In Figure 6, GOF18 EpiS cells are shown to convert to naïve pluripotency in the presence of 2i+LIF but not 2i alone, and this correlates with an enhanced induction of Tcfp2l1. In contrast, 2i+LIF (or 2i) is not sufficient to induce reprogramming of O4GIP-EpiS cells, which is however elicited by either the hyper-activation of endogenous Stat3 or Tcfp2l1 forced expression in the presence of 2i. Could the authors verify whether (or not) both GOF18 and O4GIP-EpiS cell lines express LIF receptor? Variability in EpiS cell reprogramming efficiency could also reflect the initial transcriptional status of Tcfp2l1 itself in Activin + Fgf conditions; repressed in O4GIP cells, but expressed either at low levels or within a subpopulation of GOF18 cells (to be assessed by western blotting and/or IF), yet further induced in response to 2i+LIF. Clarifying this point might further support the view that Tcfp2l1 expression closely delineates the transition between naïve and primed pluripotency states, as previously suggested (Pelton et al., 2002).

Minor concerns:

- Figure 1B and Figure 2D: please define RPKM in legend figure.
- Figure 3A: could the authors validate that all five PB-transgenes are expressed at similar levels (western blotting) prior to and after LIF removal?
- Figure 3: Data showing that Tcfp2l1-depleted ES cells are rescued in 2i + LIF need to be included in this manuscript.
- Figure 4: Different panels use 2i+LIF instead of PD +LIF as control conditions to be compared with PD alone. This needs to be changed or appropriately justified.
- Figures 6C and 6I: Legend axes are missing. Note that data presentation in these panels does not allow direct comparisons in gene expression levels between O4GIP and GOF18 cells in Activin + Fgf conditions.
- Figure 6K: please include ES cells 2i + LIF control as in Figure 6G and comment on any potential difference in the extent to which the expression of ES cell-associated factors is regained in O4GIP and GOF18 cells upon Tcfp2l1 forced expression.

Referee #3:

The authors use a genome-wide approach and compared the transcriptome of Stat3-null ESCs with available transcriptome data to identify genes, whose expression could sustain self-renewal and pluripotency in the absence of LIF. They found a transcription factor, Tfcp2l1, which is necessary and sufficient to maintain pluripotency and self-renewal of ESCs and essential for the reprogramming of EpiSCs. The authors concluded Tfcp2l1 to be a major component of the LIF/Stat3 signaling pathway that is linked to the core transcription factors required for the maintenance of naive pluripotency.

Overall the data are convincing and in accordance with the author's conclusions. However, there are minor comments to be addressed.

1) It is surprising that the authors did not find NANOG in their screening. NANOG shown to support LIF-independent self-renewal of mouse ESCs (Niwa et al., 2009. *Nature* 460: 118-122). Additionally, the authors have previously shown that the pluripotent factor Esrrb was also necessary and sufficient to sustain self-renewal and pluripotency of ESCs downstream Gsk3 inhibition (Martello et al., 2012. *Cell Stem Cell* 11: 491-504). Thus, NANOG, Esrrb and Tfcp2l1 are downstream of the LIF/Stat3 signaling pathway and are sufficient to maintain self-renewal and pluripotency of ESCs. Is the function of these pluripotent-promoting factors independent of one another? How is the role of Tfcp2l1 in sustaining pluripotency and self-renewal integrated in a more global context based on what is already known? This could highlight the relevance of Tfcp2l1 in ESC pluripotency.

2) In Figures 2E and 2G, the authors found Gbx2 expression to be responsive yet independent of LIF and suggested the potential of additional mechanisms associated to Gbx2 expression in ESCs. These data are somewhat divergent from recent findings showing Gbx2 as a LIF/Stat3 downstream target, which upon forced expression is sufficient to maintain self-renewal of ESCs in the absence of LIF. (Tai and Ying, 2013. *J. Cell Sci.* 126: 1093-1098). Furthermore, ectopic expression of Gbx2 enhanced reprogramming of MEFs into iPS and was sufficient to reprogram EpiSCs to pluripotent state ESCs (Tai and Ying, 2013. *J. Cell Sci.* 126: 1093-1098). Again, how is Tfcp2l1 function integrated with that of Gbx2?

3) Regarding Figure 4E, it would be best to include a graphical representation to better depict the result in the colony formation assay.

4) The heatmap on Figure 6C shows the expression of Tfcp2l1 along with Gbx2 and Socs3 to be significantly increased after 48 hrs in 2i media under activated Stat3 conditions. Interestingly, the role of Socs3 was not evaluated by siRNA-targeting in the reprogramming assay on Figure 6D. Additionally, the effect of knocking down Pim1 does not seem to be statistically significantly different (by looking at the error bars) from the siRNA-mediated down regulation of Tfcp2l1.

5) A schematic representation or a model figure by incorporating the role of Tfcp2l1 into what is already known about the maintenance of pluripotency and self-renewal could greatly benefit the understanding of the data.

6) Consider changing the title to: a) Identification of a new Pluripotency Promoting Factor Downstream of Leukaemia Inhibitory Factor; or b) Identification of Tfcp2l1 as a novel Pluripotency Promoting Factor Downstream of Leukaemia Inhibitory Factor

1st Revision - authors' response

11 July 2013

Referee #1:

The paper by Martello et al, "Identification of the critical pluripotency factor downstream of LIF" describes the identification of Tfcp2l1 as a target of Stat3 regulation and a key mediator of LIF signalling that maintains ES cell pluripotency.

This is a very nice study. The authors logically analysed transcriptome data to identify plausible candidates of LIF/Stat3 signalling. They proceeded to perform the necessary experiments, perturbing Tfcp2l1 expression upward and downward, to establish the role of this transcription factor. The paper is well written, the experiments were well executed and the data interpretation was sound. It was a pleasure to review this paper.

This reviewer only offers a few minor comments that would improve the paper.

1. The title is a bit grandiose in the stating "the critical pluripotency factor." At least say "a critical pluripotency factor." Perhaps even better to simply state that Tfcp2l1 is the factor.

The point of this study was to identify the essential mediator(s) of the ES cell self-renewal response to LIF. The results clearly demonstrate that Tfcp2l1 is paramount amongst Stat3 targets. However, it is not the only target and therefore it would be incorrect to describe as "the factor". We have therefore changed the title to "Identification of the Missing Pluripotency Factor Downstream of Leukaemia Inhibitory Factor". We consider this is an appropriate reflection of our findings and their significance for the field

2. The data figures are rather packed. Perhaps some information could be moved to supplementary section. For example, Figure 1 mostly reiterates what was already published in Ying et al 2008, so might be better as supplementary data.

We accept the referee's comment and have removed much of Figure 6 to the supplementary section. However, we have retained Figure 1 because these data significantly extend previous reports by showing for the first time that Stat3 -/- cells have full ES cell developmental capacity and are very similar to other ES cells in expression of a large set of pluripotency and lineage specific markers.

3. The 6136 genes expressed in ES cells and bound by Tfcp2l1 should be included as supplementary data.

We have removed Figure 5F and the text related to it (see point 4). The list of genes bound by Tfcp2l1 is now in Table S5.

4. In results section, it is not clear how it is concluded that "This widespread binding suggests that Tfcp2l1 may be a rather general regulator..." Please clarify, move to discussion, or remove.

We have removed this comment

5. Legend in Figure 3 has a typo: written as "3E", but should be "3H".

Corrected.

6. Scale bars should be added to figures 3H and 4C.

Done

7. Pim1 comes up several times in the results, but gets no mention in the discussion. Perhaps some comment is warranted.

We have added mention of Pim1 in the Discussion.

8. Be good to comment about whether other pluripotency TF's bind at Tfcp2l1.

This is a good point and is now included.

9. Please provide section headings in results section.

Done

Referee #2:

In this manuscript, Graziano Martello and colleagues uncover Tfcp2l1 as a critical downstream target of LIF/Stat3 signalling pathway in mouse ES cells. Here, the authors take advantage of Stat3^{-/-} ES cells, which retain their self-renewal and pluripotent abilities if maintained in the presence of

Mek and Gsk3 (2i) inhibitors. Other conditions (serum+LIF and PD+LIF) in contrast severely compromise Stat3^{-/-} ES cell self-renewal, as the cells fail to respond to LIF signal transduction. Combining RNA-Seq analysis in LIF-stimulated Stat3^{+/+} and Stat3^{-/-} ES cells and Stat3 ChIP-Seq dataset, they identify and validate Tcfp211 as a primary LIF/Stat3 target. Accordingly, Tcfp211 forced expression can substitute for LIF/Stat3 in Stat3^{-/-} ES cells and furthermore supports an undifferentiated ES cell state in non-permissive conditions. Conversely, conditional loss of Tcfp211 is detrimental to ES cells unless grown in 2i+LIF conditions. Interrogation of Tcfp211 ChIP-Seq dataset indicates that Tcfp211 might partly act 'outside' the core transcriptional network to sustain ES cell self-renewal, with many Tcfp211-bound target genes being expressed in 2i+LIF conditions. Beyond its role in maintaining the ES cell identity, Tcfp211 is also revealed as a potent mediator of EpiS cell reprogramming to naïve pluripotency. Overall, I believe that this manuscript would be a nice candidate for EMBO and would be of interest to a broad readership in the fields of transcription, pluripotency and reprogramming.

Major concerns:

1- The study indicates that Tcfp211 potentially functions in parallel to the core transcriptional network to sustain ES cell self-renewal but falls short in explaining how Tcfp211 operates in this context.

Figure 3F-G suggests that Tcfp211 plays a role in ES cell survival in addition to restraining differentiation. Could the authors clarify whether Tcfp211 conditional deletion indeed leads to an increased incidence of apoptosis (potentially rescued in the presence of caspase inhibitors) and whether "surviving" colonies retain a positive AP-staining?

Important clues might also come from detailed gene ontology analysis of Tcfp211-bound targets (Figure 5E) with a specific focus on Tcfp211-bound genes that are expressed in 2i+LIF conditions as compared to serum + LIF (Marks et al. 2012).

We should clarify that the cell death we described following Tcfp211 knockdown in 2i is attributable to the culture conditions. The Mek inhibitor PD0325901 causes cell death in most differentiating cell types. We observe similar levels of cell death and differentiation when non-manipulated ES cells are exposed to PD alone in N2B27 media for more than 4 days. We repeated the experiments of Figure 3F-G without the MEK inhibitor. This shows that Tcfp211 knockdown results in accelerated ES cell differentiation with no increase in cell death. We have replaced the previous figure panels which are moved to Supplementary. Similar results were obtained in LIF+serum conditions but are not included due to space constraints.

To further address the Referee's concern we have:

- *included data showing that Tcfp211 knockdown in 2i+LIF does not compromise self-renewal or survival, either at clonal density or in bulk culture (Figure S1D-E);*
- *performed Gene Ontology analysis of Tcfp211 target genes in LIF+serum (see Figure S3A). We found only 6 GO terms that barely reached statistical significance; they are related to glucose metabolism (Table below).*

2- The authors report that Tcfp211 is homogeneously expressed in ES cells in contrast to Klf4 and many other pluripotency transcription factors (Figure 3H), suggesting that Tcfp211 "... may underlie

the capacity of LIF to sustain self-renewal efficiently in serum when other factors are more restricted". This is an interesting observation that needs to be further explored, as this could provide additional insights into how *Tfcp2l1* operates in ES cells. The authors should perhaps take advantage of the recently established *Esrrb*^{-/-} ES cells (Martello et al. 2012), as these cells were shown to strictly rely on LIF to accommodate the loss of *Esrrb*.

To obtain additional insights into how Tfcp2l1 operates in ES cells in unbiased fashion we identified genes that are both bound by Tfcp2l1 and whose expression is affected by changes in Tfcp2l1 levels. We used publicly available microarray data from ES cells where Tfcp2l1 was either overexpressed or knocked-down (from Nishiyama et al. 2013 and Correa Cerro et al., 2011) and generated lists of responsive genes. We then intersected with Tfcp2l1 bound genes (from the ES cell ChIP-seq Compendium) and obtained 156 putative direct targets of Tfcp2l1 (Figure S3A). Among the putative targets we found three transcription factors that were previously described as key components of the pluripotency network, namely: Esrrb, Nanog and Tbx3 (Figure S3B).

These 3 genes were present in all gene expression analysis experiments (Figure 3G, 4F, 6G) and re-inspection of these data revealed that Tbx3 and Nanog correlate with Tfcp2l1. Collectively these data indicate that Tfcp2l1 engages with the pluripotency network through up-regulation of Nanog and Tbx3.

With specific regard to Esrrb^{-/-} cells, in LIF+serum they show reduced levels of Klf4 and Tbx3, but normal levels of Nanog and Tfcp2l1 (Figure 4F - Martello et al., 2012 and data not shown), suggesting that in this context Tfcp2l1 maintains self-renewal mainly through activation of Nanog. It should be noted, however, that Esrrb null ES cell self-renewal efficiency is not identical to wildtype ES cells.

3- In Figure 6, GOF18 EpiS cells are shown to convert to naïve pluripotency in the presence of 2i+ LIF but not 2i alone, and this correlates with an enhanced induction of *Tfcp2l1*. In contrast, 2i+ LIF (or 2i) is not sufficient to induce reprogramming of O4GIP-EpiS cells, which is however elicited by either the hyper-activation of endogenous Stat3 or *Tfcp2l1* forced expression in the presence of 2i. Could the authors verify whether (or not) both GOF18 and O4GIP-EpiS cell lines express LIF receptor? Variability in EpiS cell reprogramming efficiency could also reflect the initial transcriptional status of *Tfcp2l1* itself in Activin + Fgf conditions; repressed in O4GIP cells, but expressed either at low levels or within a subpopulation of GOF18 cells (to be assessed by western blotting and/or IF), yet further induced in response to 2i+LIF. Clarifying this point might further support the view that *Tfcp2l1* expression closely delineates the transition between naïve and primed pluripotency states, as previously suggested (Pelton et al., 2002).

The Referee here suggests three alternative explanations for the spontaneous conversion to naïve pluripotency observed in GOF18 cells, namely:

- a) differences in the initial expression levels of Tfcp2l1 at the population level;*
- b) expression of Tfcp2l1 in a subpopulation of GOF18 cells;*
- c) differential expression of LIF/Stat3 pathway components.*

We tested all 3 hypotheses:

a) we directly compared the mRNA levels of Tfcp2l1 in GOF18 and O4GIP EpiSC cells in F/A and found no significant differences (Figure S2F). Of note, Tfcp2l1 expression in both EpiSC lines is barely detectable, ~50 fold lower than in ES cell levels.

b) we performed Oct4/Tfcp2l1 immunostaining and analysed more than 1400 individual cells and, although >90% were Oct4 positive, we could not find a single Tfcp2l1 positive cell; we therefore conclude that Tfcp2l1 protein is not detectable in GOF18 cells.

c) we took advantage of publicly available microarray data from Bernemann et al., 2011, where 6 EpiSC lines and 1 mouse ES cell line were compared. Three lines (in blue in Figure S2E) were derived under the same conditions used for GOF18 cells and similarly showed spontaneous conversion to naïve pluripotency. Three other lines (in red) were generated under different conditions and did not show spontaneous conversion, similar to O4GIP EpiSCs. This analysis reveals that all six lines show comparable expression levels of LIF/Stat3 pathway components and Stat3 targets, suggesting that the ability to convert to naïve pluripotency is not simply due to differences in expression LIF/Stat3 pathway components. Unfortunately we could not compare the expression levels of Tfcp2l1 among different EpiSC lines because the relevant microarray probes did not pass the Quality Control step during analysis.

These observations are consistent with our hypothesis that spontaneous conversion to naïve

pluripotency relies on the ability to activate Tfcp2l1, not on prior expression. Tfcp2l1 activation might depend on the epigenetic status of the Tfcp2l1 locus.

- Figure 1B and Figure 2D: please define RPKM in legend figure.

Done

- Figure 3A: could the authors validate that all five PB-transgenes are expressed at similar levels (western blotting) prior to and after LIF removal?

Good antibodies for all 5 factors are not available and Western blotting is a semi-quantitative assay, therefore we used qRT-PCR to measure the expression levels of the PB-transgene. We used primer pairs detecting the total levels of each mRNA (endogenous + transgene) and compared each transgenic line to an empty-vector transfected line. This allowed us to obtain the fold change in expression over the endogenous levels for each gene. As shown in Figure S1B, the transgene expression levels are similar and range from 1.5 to 3 fold over the endogenous levels.

- Figure 3: Data showing that Tcfp2l1-depleted ES cells are rescued in 2i + LIF need to be included in this manuscript.

Done, Figure S1D-E.

- Figure 4: Different panels use 2i+LIF instead of PD +LIF as control conditions to be compared with PD alone. This needs to be changed or appropriately justified.

The main point of the western blot experiment was to show that Tfcp2l1 was expressed at nearly endogenous levels in PB-Tfcp2l1 cells. We reasoned that expression of Tfcp2l1 in LIF+PD would be lower than in 2i+LIF and therefore used PB-vector cells cultured in LIF+PD as a more stringent control. We acknowledge that this might generate confusion and therefore we repeated the Western blot using PB-Vector cells cultured in 2i+LIF and have replaced the panel (Figure 4B).

- Figures 6C and 6I: Legend axes are missing. Note that data presentation in these panels does not allow direct comparisons in gene expression levels between O4GIP and GOF18 cells in Activin + Fgf conditions.

We added the missing Legend axes. We generated a new panel where we directly compared the expression of Tfcp2l1 in O4GIP and GOF18 cells (Figure S2F). Moreover the "ES cell 2i+LIF" samples used in Figure 6C and S2B are identical, allowing direct comparison between the two experiments. This is now stated in the Figure legends.

- Figure 6K: please include ES cells 2i + LIF control as in Figure 6G and comment on any potential difference in the extent to which the expression of ES cell-associated factors is regained in O4GIP and GOF18 cells upon Tcfp2l1 forced expression.

The ES cell 2i+LIF control has been included. We observe a marked increase in Tfcp2l1 levels in both iPS lines, probably due to the presence of the PB-transgene. The extent to which naïve pluripotency markers are regained in the two EpiSCs lines is comparable (the "ES cells 2i+LIF" control used in Figure 6G and S2D is the same, allowing direct comparison). Moreover, both lines showed an increased expression of Nanog and Tbx3, probably due to the increased expression of Tfcp2l1.

Referee #3:

The authors use a genome-wide approach and compared the transcriptome of Stat3-null ESCs with available transcriptome data to identify genes, whose expression could sustain self-renewal and pluripotency in the absence of LIF. They found a transcription factor, Tfcp2l1, which is necessary and sufficient to maintain pluripotency and self-renewal of ESCs and essential for the reprogramming of EpiSCs. The authors concluded Tfcp2l1 to be a major component of the LIF/Sat3 signalling pathway that is linked to the core transcription factors required for the maintenance of naïve pluripotency. Overall the data are convincing and in accordance with the author's conclusions.

However, there are minor comments to be addressed.

1) It is surprising that the authors did not find NANOG in their screening. NANOG shown to support LIF-independent self-renewal of mouse ESCs (Niwa et al., 2009. Nature 460: 118-122). Additionally, the authors have previously shown that the pluripotent factor Esrrb was also necessary and sufficient to sustain self-renewal and pluripotency of ESCs downstream Gsk3 inhibition (Martello et al., 2012. Cell Stem Cell 11: 491-504). Thus, NANOG, Esrrb and Tfcp2l1 are downstream of the LIF/Stat3 signalling pathway and are sufficient to maintain self-renewal and pluripotency of ESCs. Is the function of these pluripotent-promoting factors independent of one another? How is the role of Tfcp2l1 in sustaining pluripotency and self-renewal integrated in a more global context based on what is already known? This could highlight the relevance of Tfcp2l1 in ESC pluripotency.

We do not understand the referee's surprise since there has never been any reliable evidence that either Nanog nor Esrrb lie directly downstream of LIF. We agree, however, that integration of Tfcp2l1 with the pluripotency network is an important point. We addressed this through an unbiased search for Tfcp2l1 targets (See Point 2 of Referee 2). We identified Nanog and Tbx3 as likely downstream effectors of Tfcp2l1.

2) In Figures 2E and 2G, the authors found Gbx2 expression to be responsive yet independent of LIF and suggested the potential of additional mechanisms associated to Gbx2 expression in ESCs. These data are somewhat divergent from recent findings showing Gbx2 as a LIF/Stat3 downstream target, which upon forced expression is sufficient to maintain self-renewal of ESCs in the absence of LIF. (Tai and Ying, 2013. J. Cell Sci. 126: 1093-1098). Furthermore, ectopic expression of Gbx2 enhanced reprogramming of MEFs into iPS and was sufficient to reprogram EpiSCs to pluripotent state ESCs (Tai and Ying, 2013. J. Cell Sci. 126: 1093-1098). Again, how is Tfcp2l1 function integrated with that of Gbx2?

Tfcp2l1 and Gbx2 are both Stat3 direct targets and, as the Referee pointed out, are both able to confer LIF-independent self-renewal and to reprogram EpiSCs. The regulation of the 2 genes is different, however: when ES cells exit the pluripotent state Tfcp2l1 is completely downregulated within the first 24h, whereas Gbx2 expression is maintained for >36h (Figure 2G and data not shown, see also Figure S2 in Ivanova et al., 2006 for a direct comparison). These results are consistent with data reported by Tai and Ying, where the expression of Gbx2 was assayed only after 48h of EB differentiation. Concerning the integration of Tfcp2l1 and Gbx2 functions, we agree on the fact that overexpression of both factors confer LIF-independent self-renewal, but not with the same potency because only Tfcp2l1 promotes self-renewal when forced expression is constrained to endogenous levels (See Figure 4B and S1B). Most importantly, downregulation of Gbx2 did not compromise self-renewal (Figure 3D and Tai and Ying 2013), whereas Tfcp2l1 knockdown resulted in ES cell differentiation. In addition, Gbx2 and Tfcp2l1 are both induced during EpiSC reprogramming triggered by LIF, but only Tfcp2l1 is required during the process (Figure 6D). We conclude that Gbx2 and Tfcp2l1 have partially redundant activities, but Tfcp2l1 is the main mediator of LIF/Stat3 activity. To clarify these points we have introduced a summary schematic in Figure 6

3) Regarding Figure 4E, it would be best to include a graphical representation to better depict the result in the colony formation assay.

A graphical representation has been provided.

4) The heatmap on Figure 6C shows the expression of Tfcp2l1 along with Gbx2 and Socs3 to be significantly increased after 48 hrs in 2i media under activated Stat3 conditions. Interestingly, the role of Socs3 was not evaluated by siRNA-targeting in the reprogramming assay on Figure 6D. Additionally, the effect of knocking down Pim1 does not seem to be statistically significantly different (by looking at the error bars) from the siRNA-mediated down regulation of Tfcp2l1.

Socs3 is a known direct target of the Jak/Stat pathway and has been used here as a positive control for activity of the pathway. Socs3 is not a transcription factor but a negative regulator of JAK signalling and its knockdown is expected increase the reprogramming efficiency. For these reasons we have not evaluated the role of Socs3 in the reprogramming assay. Concerning the effect of Pim1, siPim1 effects were variable between experiments, as shown by the

error bar. We performed a statistical test and obtained a P-value <0.01 for siStat3 and siTfcp211 and a P-value of 0.274 for siPim1.

5) A schematic representation or a model figure by incorporating the role of Tfcp211 into what is already known about the maintenance of pluripotency and self-renewal could greatly benefit the understanding of the data.

We now provide a schematic in Figure 6 illustrating how Tfcp211, and other Stat3 targets, engage with the pluripotency network.

6) Consider changing the title to: a) Identification of a new Pluripotency Promoting Factor Downstream of Leukaemia Inhibitory Factor; or b) Identification of Tfcp211 as a novel Pluripotency Promoting Factor Downstream of Leukaemia Inhibitory Factor

Since Tfcp211 is a known gene it is not appropriate to describe as novel. We have therefore used "missing" to denote its previously unattributed function.